# Irrigation return flow causing a nitrate hot spot and denitrification imprints in groundwater at Tinwald, New Zealand

Michael Kilgour Stewart[1], Philippa Lauren Aitchison-Earl[2]

[1]Aquifer Dynamics and GNS Science, PO Box 30 368, Lower Hutt 5040, New Zealand
[2]Environment Canterbury, PO Box 345, Christchurch 8011, New Zealand

*Correspondence to*: Michael K. Stewart (m.stewart@gns.cri.nz)

**Abstract.** Tinwald groundwater exhibits two features stemming from irrigation with local groundwater (i.e. irrigation return flow). The first is increased concentrations of nitrate (and other chemicals and stable isotopes) in a 'hot spot' around Tinwald. The chemical concentrations of the groundwater are increased by recirculation of water already relatively high in chemicals. The irrigation return flow coefficient C (irrigation return flow/irrigation flow) is found to be consistent with the chemical enrichments. The stable isotopes of the groundwater show a similar pattern of enrichment by irrigation return flow of up to 40% and are also enriched by evaporation (causing loss of about 20% of the original water mass). Management implications are that irrigation return flow needs to be taken into account in modelling of nitrate transport through soil/groundwater systems and in avoiding overuse of nitrate fertilizer leading to greater leaching of nitrate to the groundwater and unnecessary economic cost. The second feature is the presence of 'denitrification imprints' (shown by enrichment of the $\delta^{15}N$ and $\delta^{18}O_{NO3}$ values of nitrate) in even relatively oxic groundwaters. The denitrification imprints can be clearly seen because (apart from denitrification) the nitrate has a blended isotopic composition due to irrigation return flow and N being retained in the soil-plant system as organic-N. The nitrate concentration and isotopic compositions of nitrate are found to be correlated with dissolved oxygen concentration. This denitrification imprint is attributed to localised denitrification in fine pores or small-scale physical heterogeneity where conditions are reducing. The implication is that denitrification could be occurring where it is not expected because groundwater DO concentrations are not low.

## 1 Introduction

Excessive nitrate concentrations in groundwater are of great concern for human health and the environment. Such concentrations result from land use activities, including pastoral and arable farming, market gardening, application of nitrogenous fertilizers and industrial and sewage waste disposal (Laegreid et al., 1999). Pastoral farming has increased rapidly in recent years world-wide and especially in New Zealand. For example, dairy farming acreage on the Canterbury Plains in New Zealand (Fig. 1a) increased from 20,000 to 190,000 ha between 1990 and 2009 (Pangborn and Woodford, 2011). Arable farming has also increased and intensified. Health concerns of nitrate concentrations in drinking water relate to infant methaemoglobinaemia (WHO, 2016). The WHO guideline for maximum allowable nitrate concentration in freshwater is 50 mg/L or 11.3 mg/L of nitrate-N (WHO, 2017). Nitrate can also be toxic for aquatic life in groundwater-fed streams at lower levels, the New Zealand Government has set a maximum median of 30.5 mg/L or 6.9 mg/L of nitrate-N for river systems. Hereafter in this paper 'nitrate' is quantified as concentrations of nitrate-N in mg/L.





Nitrate concentrations in groundwater in the Tinwald area have historically been high, commonly greater than 11.3 mg/L within an area approximately 3 km wide and 11 km long (Fig. 1b). This area of high nitrate was first identified by Hanson (2002), but nitrate was already elevated in the area in the 1980s (Aitchison-Earl, 2019). The

high values are due to the history of land use in the overall area, but the Tinwald values are accentuated because the area is irrigated with local groundwater which has relatively high nitrate concentrations, whereas surrounding areas are irrigated with alpine river water with low nitrate concentrations. The terms 'irrigation return flow' (e.g. Chakraborty et al., 2015) and 'groundwater recirculation' (Brown et al., 2011) are often applied to situations where irrigation is from water that has been pumped from the underlying aquifer. This situation is common around

the world sometimes with unrecognised effects on chemical concentrations (Sánchez Pérez et al., 2003). An important and well-recognised example of the effects of irrigation return flow is non-point sourced arsenic pollution in the groundwater of the Bengal basin, regarded as one of the largest public health concerns in human history (Edmunds et al., 2015).

Irrigation return flow has important implications for water resources management as regards understanding and

modelling of nitrate transport in groundwater systems. Much effort is being expended to model the effects of nitrate produced by farming practices in order to substantiate the introduction of appropriate controls on farming to protect the water supplies of downstream communities (e.g. Environmental Canterbury website, 2020). Irrigation return flow can seriously distort such modelling by extending the time scale of nitrate transport by abstracting water from groundwater downstream and applying it upstream, and also by adding nitrate on a second

pass through the soil. This work examines the chemical and isotopic compositions of Tinwald groundwater to look for signatures attributable to irrigation return flow and how it contributes to the nitrate hot spot at Tinwald. Similar effects are expected to be important for many other locations in agricultural areas throughout the world.

Irrigation return flow also appears to contribute to an enhanced 'denitrification Imprint' in groundwater at Tinwald. Denitrification imprints are discernible in even reasonably oxic groundwaters. The stable isotopes of

nitrate ($^{15}$N and $^{18}$O) have often been used to investigate both the sources of the nitrate and its natural attenuation via denitrification (i.e. microbial reduction of nitrate) (e.g. Mariotti et al., 1988, Böhlke and Denver, 1995, Kendall, 1998). Understanding the sources of nitrate is important for remediation of excessive nitrate concentrations as at Tinwald (Aitchison-Earl, 2019). Natural attenuation of nitrate via denitrification is a vital eco-service to the environment, and comparison of estimates of nitrate loss by leaching from the bottom of the

root zone in catchments compared with the outflow of nitrate from streams shows that considerable attenuation of nitrate occurs in the vadose zone-groundwater continuum. However, little is known about the detailed processes affecting nitrate transport and fate in this region (Clague et al., 2015, Wells et al., 2016, Stenger et al., 2018, Burbery, 2018).

In summary, the objectives of this paper are to investigate the role of irrigation return flow in:

1.    Accentuating the nitrate hot spot at Tinwald, and

2.    Producing denitrification imprints in relatively oxic groundwaters.

**2 Background**

**2.1 Geohydrologic setting**





The study area centres around the small town of Tinwald (population 3000) situated on the south bank of the
Hakatere/Ashburton River and located on the large coalescing alluvial plain known as the 'Canterbury Plains'
(Fig. 1a). The Canterbury Plains were built up by rivers fed by glaciers over several million years. Deposition in
the Tinwald area (Fig. 1b) was mainly by the South Branch Hakatere/Ashburton River and its ancestors (Barrell
et. al, 1996). The alluvial deposits are poorly stratified greywacke gravel dominated with silts and sands which
become finer towards the coast. Oil well exploration drilling and seismic surveys of the Ashburton-Hinds areas
indicate thicknesses of over 1000 m of alluvial gravels overlying marine sediments (Jongens et. al, 2012).
Existing wells In the Tinwald study area are almost all less than 100 m, and over half are less than 40 m deep
(Aitchison-Earl, 2019). Wells are generally screened within post-glacial (Holocene) or last glacial (Late
Quaternary) deposits. Shallow wells and springs are common close to the river, within the Holocene age deposits.
There is little geological impedance for movement of groundwater between shallower and deeper screened wells.
The regional groundwater flow direction is parallel to the Hakatere/Ashburton River. State Highway 1 runs
through the study area (Fig. 1b) and was originally built to take advantage of drier conditions at the inland point
of the old 'Hinds swamp'. The swamp has been largely drained but influences soil types, with deeper, poorly
drained organic soils with less leaching and greater denitrification potential located coastwards of the highway.
Soils are lighter and more freely draining with greater nitrate leaching risk inland of the highway and adjacent to
the Hakatere/Ashburton River (Landcare Research, 2015).
In the Tinwald study area, two wells with depths less than 50 m had mean ages of 12 and 63 years based on CFC
and tritium measurements (Stewart et al, 2002; van der Raaij, 2013). Ages generally increase with depth in other
wells in the greater Ashburton area. A trial site for managed aquifer recharge (MAR) has been operating since
2016 just outside of the study area (Fig. 1b).

## 2.2 Hydrology

The closest long-term rainfall site is part of a climate station at Ashburton Council (Fig. 1b). Annual average
rainfall at Ashburton Council is around 730 mm (measured between 1909-2017), ranging from 382 to 1147 mm.
There is little seasonality in rainfall, which averages 61 mm a month. Groundwater recharge was reported by
Thorpe and Scott (1999) based on lysimeter measurements of soil drainage at Winchmore (10 km north of
Ashburton, Fig. 2). In the 10-year period (1961-1971), average recharge was 293.5 mm/year with average rainfall
of 730 mm/year and PET of 765 mm/year. Average monthly recharge was much higher in winter months (April
to September). Winchmore soil is described as Lismore Stony Silt Loam characteristic of that at Tinwald west of
Highway 1, and much of the Canterbury Plains.
The Hakatere/Ashburton River has a north and south branch sourced from the Canterbury Ranges which converge
at the north of the study area. The Hakatere/Ashburton River interacts with local groundwater, losing and gaining
water along its length. Flow is lost to groundwater from the South Branch, and gains towards the confluence with
the North Branch.
Springs and wetlands indicate areas where the water table is naturally close to the surface and groundwater
discharge is occurring. Many springs are found in the Hakatere/Ashburton River catchment, and often occur in
relict river channels (Aitchison-Earl, 2000). In the study area, Carters Creek and Laghmor Creek are both sourced
from springs, and there are springs above Lake Hood that flow into the lake (Fig. 1b).

### 2.3 Land and groundwater use





Cropping has been a major land use in the Tinwald area since at least the early 1940s (Fig. 1b, Engelbrecht, 2005).
Most of the area is not part of any of the major surface water irrigation schemes, so irrigation was developed from
groundwater sources within the area from the 1980s. Cultivation and fertilizer practice in cropping has an impact
115    on the amount of nitrate that is leached from the soil to the groundwater. Winter is the most likely time for leaching
to occur due to saturated soils and less nitrogen being used by crops. Nitrogen-fixing clover crops have been used
less over time with an increase in commercial fertilizers (predominantly urea). Point sources of nitrate and other
contaminants include septic tanks (human effluent), dairy and other animal effluent, stormwater and contaminated
120    water.

Groundwater use in the Tinwald area is mainly for irrigation, and for domestic and stock water supply. The Valetta
Irrigation scheme extends to the edge of the study area in the north-west and sources water from the braided alpine
Rangitata River to the south (Figs. 2 and 4).

### 2.4 Nitrate concentrations

125    Consistently high nitrate concentrations (greater than MAV of 11.3 mg/L) in groundwater were first identified in
the Tinwald area in 2002 (Hanson, 2002). Maximum recorded nitrate concentrations from all samples between
1990 and 2017 are shown in Fig. 2. In general, lower nitrate concentrations occur close to the rivers and under
and down-gradient of surface water irrigation schemes. This is because of dilution by river-sourced water which
is lower in nitrate. Nitrate concentrations are higher in areas with more land-surface recharge.

130    Nitrate concentrations are lower towards the coast in the old Hinds swamp (southeast of SH1). The lower nitrate
concentrations are driven by reducing conditions which facilitate denitrification (Hanson and Abraham, 2010).
(Note that there are elevated nitrate levels outside the study area from the east side of Ashburton to the coast.
which are the result of historic leaching from meat processing plants. These are not part of this study.)

Nitrate concentrations have increased over time in the Tinwald area, with two long-term monitoring sites (Thews
135    Road  and Saleyards wells, Fig. 1b) having statistically significant (P< 0.05) upwards trends of 0.44 and 0.29
mg/L/year since monitoring began in 1995 (Aitchison-Earl, 2019).

### 3 Methods

### 3.1 Sampling

33 wells were sampled in the study area between 7 February and 14 March 2018 (Fig. 3). 13 of the wells had been
sampled in 2004, and others were selected to fill gaps at a range of well depths. Groundwater levels were high at
the time of sampling, related to high rainfall and river flows following a sustained period of low groundwater
levels (Aitchison-Earl, 2019). A large rainfall event of over 100 mm occurred during the sampling period on 21
February, leading to an increase in river flow.

### 3.2 Chemical measurements

Samples were analysed for Environment Canterbury's standard suite of major ions through Hills Laboratories
(Aitchison-Earl, 2019).  Field measurements included dissolved oxygen (DO), pH, conductivity, temperature and
depth to groundwater. A selection of the field quantity and ion concentration results are given in Tables 1 and 2.



The samples have been ordered from lowest to highest DO concentrations, and four groups of samples (A to D) are identified to aid discussion.


### 3.3 Water Isotopes ($\delta^{18}O$, $\delta^{2}H$)

Water samples were analysed on an Isoprime mass spectrometer; for $\delta^{18}O$ by water equilibration at 25°C using an Aquaprep device, for $\delta^{2}H$ by reduction at 1100 °C using a Eurovector Chrome HD elemental analyser. Results are reported with respect to VSMOW2. The analytical precision for this instrument is 0.2‰ for $\delta^{18}O$ and 2.0‰

for $\delta^{2}H$. Results are given in Table 1.

### 3.4 Nitrate isotopes ($\delta^{15}N$, $\delta^{18}O_{NO3}$)

Nitrate samples ($NO_3$) were converted to nitrite ($NO_2$) using cadmium, then to nitrous oxide ($N_2O$) using sodium azide in an acetic acid buffer. The $N_2O$ was then extracted from the water sample, passed through a series of chemical traps to remove $H_2O$ and $CO_2$, and cryogenically trapped under liquid nitrogen. After being cryofocused

in a second trap, the $N_2O$ passed through a GC column and into an Isoprime IRMS to determine its isotopic signature of nitrogen and oxygen. Our method is modified from McIlvin and Altabet (2005), following personal communication with Mark Altabet. Results are reported with respect to AIR for $\delta^{15}N$ and VSMOW for $\delta^{18}O$. The analytical precision for these measurements is 0.3‰ for $\delta^{15}N$ and $\delta^{18}O_{NO3}$, except for samples below 100 mg/L $NO_3$-N which may have lower precisions. Results are given in Table 1.

## 4 Results

### 4.1 Groundwater chemistry

DO concentrations in the Tinwald groundwaters range from 0.18 to 11.8 mg/L, although the majority are high and indicate relatively oxic groundwater. As noted above, the data in Tables 1 and 2 are ordered from lowest to highest DO values.

Chloride concentrations are useful to distinguish recharge sources. Chloride concentrations are highest in rainfall originating over the sea and near the coast, and generally decrease with distance inland. In particular, alpine rivers (with chloride values of 0-5 mg/L) and coastal rainfall-derived infiltration (with chloride ranging from 10-20 mg/L) can be clearly distinguished (Hayward, 2002; Stewart et al., 2002). However, chloride concentrations in the Tinwald area (Fig. 4a) are greater than expected even for coastal rainfall (most are >15 mg/L). The values are

lower (0-10 mg/L) to the northeast side of the study area near the Hakatere/Ashburton River and to the southwest. Sulphate occurs naturally in groundwater and is present in fertilizers and fungicides, and so can be an indicator of human influence when concentrations are in excess of background levels as here (see Fig. 4b). As with chloride, levels in alpine river and low-altitude rainfall infiltration are very different, but in the case of sulphate the difference is caused by the nature of additions to the soils in the respective catchments rather than the

concentrations in rainfall. Concentrations are lower on the northeast and southwest boundaries of the study area. Nitrate concentrations are shown in Fig. 4c. Nitrate concentration exceeds MAV (11.3 mg/L) in 17 of 33 wells sampled in the study area and is over 20 mg/L in four wells. The highest nitrate concentrations cluster inland of SH1 to the west and northwest of Tinwald and underlie an area of dominant cropping land use (Fig. 1b). Nitrate





is lowest on the northeast boundary of the study area (near the Hakatere/Ashburton River) where it is generally
below ½ MAV, 5.65 mg/L, and lower but still over ½ MAV on the southwest boundary.

To investigate possible irrigation return flow effects, we compare the concentrations of different solutes and
isotopes and include the effect of evaporation as indicated by the stable water isotopes (Figs. 5a, b, c). Fig. 5a
shows water $\delta^{18}O$ versus the chloride. Higher $\delta^{18}O$ correlates with higher chloride, but this is not mainly due to
evaporation (because the evaporation vector is not parallel to the trend). Instead the main influence is the source
of the recharge because both chloride and $\delta^{18}O$ are higher in land surface recharge (e.g. Group C samples) and
lower in alpine river recharge (Group D samples). There is no effect due to DO. Sample 21 shows an extra
evaporation effect.

Fig. 5b shows sulphate and chloride are well correlated especially when the low DO samples are excluded. This
trend is also due to the recharge sources (see Group C and D samples in the figure). The trend is accentuated by
evaporative enrichment (since the evaporation arrow is parallel to the trend in Fig. 5b). The low DO samples
conform to the overall trend, but are more scattered than the other samples.

Fig. 5c shows nitrate and chloride are moderately correlated due to the recharge sources (see Group C and D
samples), but there are other processes affecting the nitrate concentrations. A small evaporation effect accentuates
the trend. The low DO waters have low nitrate concentrations indicating that they have been partially denitrified,
probably within the soil (see below).

The clear message from these results is that nitrate, sulphate and chloride concentrations are increased in areas
irrigated by local groundwater compared to those irrigated by alpine river water.

### 4.2 Water Isotopes $\delta^{18}O$ and $\delta^2H$

$\delta^{18}O$ values are useful as tracers of the sources of recharge to groundwater, because rainfall recharge and rivers
from alpine catchments have different isotope ratio signatures. Scott (2014) reviewed $\delta^{18}O$ data for Canterbury
and identified isotopic signatures in the Ashburton area. The Hakatere/Ashburton River has very negative $\delta^{18}O$
with a mean of -10.1‰ for the South Branch, and -10.7‰ for the North Branch. In contrast, rainfall recharge near
the coast is less negative than -8‰, although it becomes more negative inland and is typically more negative than
-8‰ on the upper plains. The Rangitata River, the alpine source of Valetta and other irrigation schemes water,
has a mean $\delta^{18}O$ of -9.8‰ (Taylor et al., 1989).

$\delta^{18}O$ data collected in the greater Ashburton area is shown in Fig. 6. The influence of more negative Rangitata
River sourced water can be seen under the irrigation schemes. The $\delta^{18}O$ values are less negative (red and orange
dots) in the Tinwald study area. More negative values (green dots) occur on the NE and SW boundaries of the
area, which are related to recharge from the South Branch Hakatere/Ashburton River and the Valetta scheme
irrigation water. An area south of Tinwald near the coast (Eiffleton Irrigation Scheme) has less negative $\delta^{18}O$
values like those observed in the Tinwald study area for probably the same reasons (irrigation return flow).

We also sampled wells for $\delta^2H$ in 2018. Plotting $\delta^2H$ and $\delta^{18}O$ against each other frequently gives a linear
relationship following the equation

$$\delta^2H = 8.0\delta^{18}O + d. \tag{1}$$

On a global scale, d (called the deuterium excess) is equal to +10‰ on average, and the linear relationship is
known as the Global Meteoric Water Line (Craig, 1961). A New Zealand Meteoric Water line with deuterium
excess, d = +13‰ was established by Stewart and Taylor (1981), based on westerly-dominated zones of New



Zealand (North Island and western parts of the South Island). However, Scott (2014) and previous workers found that data from Canterbury fitted local meteoric lines with d = +10‰ for the Waimakariri Zone and d = +11‰ for

the Selwyn-Te Waihora Zone. Scott (2014) noted a lack of paired δ18O and δ2H data in the Ashburton area, but available evidence supports a local meteoric water line (LMWL) for Canterbury with d = +10‰ (Taylor et al., 1989; Stewart and Morgenstern, 2001; Scott, 2014; Stewart et al., 2018). This is taken as the LMWL below. Paired δ18O and δ2H data for the Tinwald study are plotted in Fig. 7. The Tinwald data plot below the LMWL and have a linear best-fit line (excluding sample 21, which has been affected by extra evaporation) given by:

$$\delta^2 H = 6.3\delta^{18}O - 7.2 \tag{2}$$

The slope of less than 8 for this line suggests that the waters have been affected by evaporation. A ratio of about 5 in the $^2$H and $^{18}$O enrichments is expected for evaporation at ambient temperatures (Stewart, 1975). It is expected that the isotopic compositions of the water would have been enriched by evaporation and/or evapotranspiration during the irrigation return flow process. Estimates of the isotopic enrichments required to explain the

displacement of the average isotopic compositions of the group from the LMWL are shown by the red arrow with slope of 5 in Fig. 7. The average isotopic composition of the samples except sample 21 (Table 1) was (-8.58, -61.2), where the bracket represents (δ18O, δ2H). The average initial composition of the samples would then have been (-9.43, -65.4), i.e. where the red arrow with a slope of 5 meets the LMWL (marked by the small red circle in Fig. 7). This degree of isotopic enrichment is obtained by evaporation of approximately 20% of the water

according to both isotopes (see the calculation in the appendix based on Stewart, 1975). Recharge would be a mix of rainfall and irrigation water which would be evaporated. The effects would be expected to be variable from well to well as observed.

In addition, the difference in the δ18O and δ2H values of Groups C and D is attributed to their different irrigation sources (local groundwater or alpine river water) as observed for the chemical compositions. Assuming that both

groups are affected by evaporation to the same extent, the difference between the groups compared to the difference between the irrigation sources will give an approximate measure of the irrigation input. The δ18O difference between Groups C and D is 0.63‰ (Table 4) and that between the sources is 1.63‰, giving irrigation input of 39%. For δ2H it is 4.1‰ compared to 10.2‰ giving 40% irrigation input. These are likely to be overestimates because Group C waters may be more affected by evaporation than Group D waters. (The

compositions of the irrigation sources are taken as local rainfall (-8.17, -58.7) and alpine river (-9.80, -68.9)).

### 4.3 Nitrate isotopes δ15N and δ18O$_{NO3}$

The nitrate isotope results are given in Table 1 and plotted in Fig. 8a. The samples have symbols depending on their DO concentrations, as in previous figures. The figure displays two important features:

1.    There is a very good linear relationship between the δ15N and δ18O$_{NO3}$ values of the nitrate,

2.    Their positions along the line depend on their DO concentrations.

The first feature is the linear relationship between the δ15N and δ18O$_{NO3}$ values of the nitrate (except sample 06 and to a smaller extent samples 02 and 03). Denitrification causes increased δ values of

nitrate, along with decrease of nitrate concentration. The slope of the isotopic enrichments caused by denitrification (i.e. enrichment in δ18O$_{NO3}$/enrichment in δ15N) has been reported to be in the range 0.48 – 0.77

(Kendall, 1998; Burns et al., 2011; Kaushal et al., 2011; Zhang et al., 2019). The line shown in Fig. 8a has a slope of 0.68 and was calculated to simulate the effect of denitrification using the Rayleigh formula to represent the





process (Kendall, 1998). Similar denitrification line slopes of 0.73 and 0.75 were observed by Clague et al. (2015) and Stenger et al. (2018) respectively. The individual denitrification calculations for each of the isotopes are plotted in Figs. 8b and c (see explanation below). The starting point for the denitrification lines was chosen to be

the average of the Group C samples. The linear relationship shows that either the various sources of nitrate all produce nitrate with the same isotopic δ-values (which is contrary to what we know) or more probably nitrate leaching from the soil is blended by processes in the soil (Wells et al., 2015) and by irrigation return flow. The exceptions are sample 06 and to a lesser extent samples 02 and 03, their isotopic compositions (Fig. 8a) suggest that they initially had higher $\delta^{15}N$ than the other samples and therefore a greater proportion of effluent nitrate.

The second feature of the figure is surprising. Denitrification is only expected to take place where DO levels are very low (e.g. < 0.5 mg/L, McMahon and Chapelle, 2008). But here denitrification effects are observed when the DO concentrations in the groundwaters are much higher. This must mean that the denitrification occurred during the past histories of the nitrates, most probably within the soil the nitrate was leached from. The lowest DO range in Table 1 (with DO < 4 mg/L, samples 01-09) includes Group A waters which have the lowest DOs and most

marked denitrification effects (with $\delta^{15}N$ values from 15 to 20‰), Group B waters with DO from 1.36 to 3.39 mg/L and $\delta^{15}N$ values from 7 to 9‰, and one other well (07) that in contrast shows only minor denitrification ($\delta^{15}N$ is 5‰) despite its relatively low DO (2.68 mg/L). The intermediate DO group (4 < DO < 8.2 mg/L) has intermediate nitrate concentrations and shows smaller denitrification effects ($\delta^{15}N$ values from 3.5 to 6.0‰, samples 10-14). The highest DO group (with DO > 8.2 mg/L) are nearly saturated with oxygen and show minimal

denitrification effects ($\delta^{15}N$ values from 1.7 to 4.8‰, samples 15-33).

Fig. 8b shows the natural log of the nitrate concentrations versus $\delta^{15}N$ values, the natural log is used because the denitrification line will be linear on this type of plot according to the Rayleigh formula. The grey bands show approximate values of the $\delta^{15}N$ values of possible nitrate sources (i.e. natural soil with $\delta^{15}N$ of -3 to 7‰, inorganic fertilizer with -3 to 3‰, and effluent with 9 to 25‰, Fogg et al., 1998; Stewart et al., 2011). The Ln ($NO_3$) values

they are plotted at are schematic, we normally expect background nitrate concentrations from natural sources in soil to be about < 0.7 to 1 mg/L (Close et al., 2001; Daughney and Reeves, 2005). Nitrogen-fixing clover is a possible source of nitrate with an isotopic composition like that of soil nitrate in pasture in Canterbury (Trevis, 2012). But we think its contribution is not large, because the abundance of clover has decreased over the years as fertilizer use (particularly urea) has increased.

The approximate Rayleigh formula used is

$$\delta = \delta_o + \varepsilon.Ln(f) \tag{3}$$

where δ is the $\delta^{15}N$ or $\delta^{18}O$ value of the nitrate remaining after the microbes have catalysed partial denitrification, and $\delta_o$ is the initial composition of the nitrate. ε is the enrichment factor for the reaction and f is the fraction of nitrate remaining after the reaction. Results of the calculation are given in Table 3.

The enrichment factors producing the denitrification lines in Figs. 8b and c are $\varepsilon(^{15}N)$ = -3.0‰, $\varepsilon(^{18}O)$ = -2.1‰. These are similar to the ranges determined by Clague et al. (2015) ($\varepsilon(^{15}N)$ = -1.1 to -9.6‰, $\varepsilon(^{18}O)$ = -1.0 to -7.2‰), and values by Stenger et al. (2018) ($\varepsilon(^{15}N)$ = -2.0‰, $\varepsilon(^{18}O)$ = -1.3‰), while Mariotti et al. (1988) gave an $\varepsilon(^{15}N)$ range from -5 to -8‰. Other authors (Kendall, 1998, and references therein) gave much larger negative values. Mariotti et al. (1988) suggested that low values may occur if denitrification occurs in dead-end pores causing a

non-fractionating sink for nitrate by diffusion. Stenger et al. (2018) considered that small-scale physical





heterogeneity, including the localised distribution of resident electron donors and the effect of lateral flows, was a more likely cause with their coarse-textured ignimbrite materials.

Fig. 8c shows the natural log of the nitrate concentrations versus $\delta^{18}O_{NO3}$ values. As with Figs. 8a and b, the denitrification line through Groups C and B wells project to sample 01. This well is located south of Tinwald near

Lagmhor Creek in the Hinds swamp denitrification area. Group A wells show the greatest denitrification effects, the other samples in the group (02, 03 and 06) lie to the right of the denitrification line in Fig. 8b indicating that they have larger proportions of effluent than the rest of the samples. Samples 02 and 03 occur downgradient of the old Tinwald Saleyards, sample 06 is northwest of SH1 in an area of lifestyle blocks adjacent to the major cropping area. It is probable that the effluent source is providing a source of dissolved organic carbon to fuel

denitrification reactions in Group A wells.

Group B wells (showing moderate denitrification) are located closest to the Hakatere/Ashburton River. One sample (04) is in the cropping area, the others (05, 08, 09) are in areas with lifestyle blocks, which could contribute both septic tank and animal effluent to assist denitrification.

Group C wells (representative of wells irrigated by local groundwater) are distributed through the central part of

the high nitrate hot spot. They plot in the upper part of the cluster of green points in Fig. 8a, and to the right in Figs. 8b and c.

Group D wells (representative of wells irrigated more by alpine river water) are located on the southwest boundary of the study area. They plot in the lower part of the green point cluster in Fig. 8a, and to the left in Figs. 8b and c. The green points in Fig. 8 have $\delta^{15}N$ values that are mostly within the soil nitrate or inorganic fertilizer ranges and

show little evidence of denitrification. Natural soil nitrate alone does not account for the elevated nitrate concentrations in these wells, making inorganic fertilizer (or rather organic-N derived from it, see discussion below) the likely dominant source.

A mixing curve between two nitrate source end members (soil and fertilizer/effluent mixture) has been fitted to the solid green points (not shown). The equation of the curve (Kendall, 1998) is:

$$\delta^{15}N = b - a/C_N \tag{4}$$

where $a$ describes the shape of the curve, $b$ is the $\delta^{15}N$ or $\delta^{18}O_{NO3}$ value of the total (blended) nitrate source, and $C_N$ is the nitrate concentration. The best-fitting curves give $\delta^{15}N = 3.6‰$, $\delta^{18}O_{NO3} = -0.2‰$ for the blended nitrate source. This indicates that the source is dominated by inorganic fertilizer but has a small proportion of effluent source based on $^{15}N$ (it is assumed that the highest nitrate concentrations are little affected by denitrification).

**5 Discussion**

**5.1 Irrigation return flow effects on chemical and isotopic concentrations**

Fertilizers have been applied to much of the area between the Ashburton and Hinds Rivers not just to the Tinwald study area, and rainfall applies to the whole area with contours of the δ values in rainfall decreasing inland from the coast (Stewart et al., 2002). Yet the Tinwald area shows elevated nitrate concentrations (and chloride, sulphate,

etc.) compared to the surrounding areas (see Fig. 2). The difference is that the Tinwald study area is irrigated by groundwater from shallow local wells, whereas much of the rest of the area is irrigated by alpine river water with low concentrations of solutes. This has affected the chemical and water isotope concentrations.





The irrigation return flow process is illustrated schematically in Fig. 9. The average chemical concentrations of Groups C and D are taken as representative of the Tinwald hot spot and outside groundwaters respectively (values

are given in Table 4). Recharge to groundwater (RGW) is given by

$$RGW = R + I - AET \qquad (5)$$

where R is rainfall, I is irrigation input and AET is actual evapotranspiration (all in mm/year). Surface runoff and lateral seepage are neglected in this treatment as both are expected to be small. (Note that drains in the area are fed mainly by groundwater.) Thorpe and Scott (1999) reported on many years of lysimeter measurements of

recharge at the nearby research station of Winchmore. They found that

$$RGW = 0.80(R + I) - 380 \qquad (6)$$

for pasture, while for bare soil it was

$$RGW = 0.77(R + I) - 244 \qquad (7)$$

The recharge fraction (RGW%) is the ratio of recharge to incident rainfall plus irrigation applied, i.e.

$$RGW\% = RGW/(R + I) \qquad (8)$$

Dewandel et al. (2007) defined an irrigation return flow coefficient C equal to the recharge from irrigation (i.e. irrigation return flow, IRF) divided by the irrigation flow itself (I). C is the same as the recharge %, i.e.

$$C = \frac{IRF}{I} = \frac{I \times RGW\%}{I} = RGW\% \qquad (9)$$

if the recharge rate is assumed to be the same for both rainfall and irrigation. C is used to quantify the effect of

irrigation return flow on the water balance.

Equations (6) and (7) can be used to determine the coefficient C for Tinwald. Assuming that I = 200 mm, C is 364/930 = 39% for pasture, and 51% for bare soil. These give predicted chemical enrichment factors (1/C) of 2.6 for pasture and 2.0 for bare soil (if chemicals put in via rainfall and irrigation are concentrated into the recharge fraction, i.e. are concentrated by the loss of AET). The pasture value may be the most appropriate to compare with

the observed values in Table 4 for the different chemicals. The observed values are very scattered, but are similar on average to the predicted value of 2.6. Chloride is the most conservative element and its enrichment factor (3.2) is a little higher than the predicted value. The sulphate enrichment factor is very large suggesting greater fertilizer input into the Group C area soil than into the Group D area soil. The bicarbonate factor is small perhaps because of chemical re-equilibration as groundwater passes through the soil in the Group C area.

The $\delta^{18}O$ and $\delta^{2}H$ values of Groups C and D are affected by the different irrigation water sources (local groundwater or alpine river water) and by evaporation as described in Section 4.2. An irrigation input of up to 40% is indicated by the mean isotopic compositions of groups C and D. Evaporation is indicated by displacement of the sample points from the LMWL in Fig. 7. An approximate calculation given in the appendix shows that evaporation of about 20% of the water can explain the average displacement of the points.

Irrigation return flow has important implications for management of nitrate in agriculture. An important aspect of water resources management is understanding and modelling of nitrate transport in water systems (in this case groundwater). Much effort is being expended to model the effects of nitrate produced by farming practices in order to introduce and substantiate appropriate controls on farming to protect the water supplies of downstream communities (e.g. Environmental Canterbury website, 2020). Irrigation return flow can seriously distort such

modelling by lengthening the time scale of nitrate transport by abstracting water from groundwater downstream and applying it upstream and by adding nitrate on a second pass through the soil.





Another effect of irrigation return flow is distortion of tracer age dating results. Tritium concentrations will not be reset by interaction with the atmosphere when irrigation water is applied to the soil, so the tritium ages of groundwater affected by irrigation return flow will appear to be older than they really are. In contrast , CFC/SF$_6$

ages will be reset to zero in the soil and groundwater ages will reflect time since recharge. This appears to be the case for data in the Tinwald area, although data is scarce (Stewart et al., 2002).

A practical consideration is that if irrigation water already contains nitrate then too much fertilizer could be applied leading to unnecessary economic cost and greater nitrate leaching potential, if the nitrate in the groundwater is not accounted for by nutrient budgeting (e.g. Flintoft, 2015).

**5.2 Nitrate dual isotope concentrations**

**5.2.1 Nitrate source identification**

Nitrate isotope results that have not been affected by denitrification (i.e. usually the oxic samples) potentially give information on the nitrate sources and also on the starting points for denitrification vectors. Numerous studies of the $\delta^{15}N$ values produced by different nitrate sources have identified ranges which have differed under local

conditions (e.g. Kendall, 1998; Fogg et al., 1998; Stewart et al., 2011, Fig. 8b). Results for oxic samples from recent New Zealand studies are given in Fig. 10. The rectangles show source signature fields resulting from urea fertilizer/soil N/ruminant excreta at Toenepi Catchment (Clague et al., 2015), urine/urea/soil N at Harts Creek (Wells et al., 2016), low intensity animal grazing (soil N/manure) at Waihora wellfield (Stenger et al., 2018), two sources (inorganic fertilisers/manure and piggery effluent) at Waimea Plains (Stewart, 2011), and inorganic

fertiliser/urea/manure at Tinwald (Groups C and D, this work). Despite the variety of nitrate sources, the $\delta^{15}N$ values generally show overlapping ranges as illustrated in Fig. 10 (except for the Waimea Plains piggery effluent source).

Use of $\delta^{18}O_{NO3}$ in combination with $\delta^{15}N$ to identify nitrate sources has not been very successful, as illustrated in Fig. 10 where the $\delta^{18}O_{NO3}$ values overlap each other. On the other hand, the combination has proven to be effective

for detecting the occurrence of processes in the nitrogen cycle, such as nitrification and denitrification (Aravena and Robertson, 1998). The only distinctive source $\delta^{18}O_{NO3}$ values are those expected for nitrate fertilizer (see 'nitrate fertilizers' box in Fig. 10, Xue et al., 2009, Wells et al., 2015). Many researchers have looked for such $\delta^{18}O_{NO3}$ values and generally failed to find them (Kloppman et al., 2018). Instead the values observed in groundwaters are usually characteristic of soil nitrate or effluent (as illustrated in Fig. 10).

The probable answer to this failure to observe the expected high $\delta^{18}O_{NO3}$ values in groundwater is that inorganic fertilizer-derived nitrate is not directly and rapidly transferred to groundwater but is retained in the soil-plant system as organic-N, and only later mineralised and re-oxidised thereby becoming available for leaching to the groundwater (Somers and Savard, 2009, Wells et al., 2015, Kloppmann et al., 2018). The process of mineralisation and re-oxidation resets the $\delta^{18}O_{NO3}$ and also changes the $\delta^{15}N$. The time delays in this process can be considerable

(as much as several decades, Sebilo et al., 2013). This means that there will be a legacy of organic-N built up in the Tinwald soil from past applications of fertilizer in addition to past soil management practices. This time delay is in addition to the time delay due to the mean residence time of the groundwater. Others have previously identified the importance of organic-N in the soil (variously known as soil organic matter (SOM, Somers and Savard, 2009) or soil organic nitrogen (SON, Wells et al., 2015)) as the pool of nitrogen within the soil controlling

the rate and timing of nitrate releases to groundwater. The transfer to organic-N is most efficient at times of high



microbial activity (spring/summer growth) and much less in low microbial activity (winter), when increased nitrate leaching to the groundwater is likely (Mengis et al., 2001; Somers and Savard, 2009).

The nitrate isotopes (Fig. 8a) show an unexpected blending of the isotopic compositions of the nitrate in the groundwater (and therefore the soil/vadose zone). This blending is considered to be due to irrigation return flow in conjunction with the action of organic-N in mediating and retaining N in the soil. This has allowed the denitrification process to be identified and explored in this study, and the enrichment factors for denitrification to be determined.

### 5.2.2 Denitrification imprint in oxic groundwater

The nitrate isotopes show clearly that denitrification is important in the Tinwald soil and vadose zones (Fig. 8). Firstly, the nitrate isotopes show that the nitrate sources are blended within the soil and that inorganic fertilizers are dominant with minor effluent input. Secondly, the nitrate concentration and isotopic compositions are correlated with the DO concentrations, despite most of the groundwaters having DO concentrations greater than the levels at which denitrification can occur (McMahon and Chapelle, 2008; Rivett et al., 2008).

The correlations are displayed in Figs. 11a-c. The relationship between DO and nitrate is approximately linear (Fig 11a) with the nitrate concentrations being more scattered at the high DO end related to the recharge sources (Groups C and D, see earlier results). The line fitted to samples with DO < 8.2 mg/L and Group C samples shows an average trend reflecting denitrification. Figs. 11b and c showing $\delta^{15}N$ and $\delta^{18}O_{NO3}$ plotted against Ln(DO) also have average linear trends fitted to them related to denitrification.

Stenger et al. (2008) pointed out a similar situation where denitrification was inferred by unexpectedly low nitrate concentrations, but DO concentrations although varied were not particularly low. There was, however, an approximate correlation of nitrate and DO, as here. Manganese (Mn) and Iron (Fe) are other indicators of reducing conditions. Both cases (Stenger et al. and Tinwald) show the expected patterns of low nitrate concentrations when Mn and Fe are elevated (indicating very reducing conditions) and higher nitrate concentrations when Mn and Fe are very low (indicating oxidising conditions).

Several factors suggest that the denitrification imprint arises from localised denitrification in fine pores where conditions are reducing. 1. Koba et al. (1997) showed that denitrification can occur in anaerobic pockets within otherwise oxic sediments or water bodies. 2. The low values of $\varepsilon(^{15}N)$ and $\varepsilon(^{18}O)$ observed here indicate that denitrification occurs in fine pores or small-scale physical heterogeneity. 3. The occurrence of the denitrification imprint in moderately oxic waters (in which denitrification could not have occurred) means that the denitrification must have occurred in parts of the system which were much more reducing. Logically these are fine pores or inhomogeneities containing electron donors with heterotrophic bacteria.

The Tinwald study area is not in an area where the groundwater is generally reducing (Close et al., 2016), but nevertheless some wells show the denitrification imprints. It would appear that denitrification imprints in moderately oxic groundwater should be common, but many more nitrate isotope measurements would be required to show this.

As a final comment, there appear to be two types of pore space in the gravels at Tinwald, i.e. larger pores with mobile water and finer pores with almost stagnant water, that communicate by diffusion. This is likely to cause slowing of nitrate transport through the system as nitrate is retained is transferred to the finer pores (together with some denitrification).



## 6 Conclusions

Chemical measurements at Tinwald corroborated previous indications of an area of high nitrate concentration in the groundwater, which partly results from irrigation return flow in the area. During the recirculation process by spray irrigation of local groundwater, the chemical composition of the groundwater is enriched by recirculation of water already relatively high in chemicals, along with enrichment by gain of chemicals from the soil and evaporation. The irrigation return flow coefficient (C) of about 0.4 indicates a chemical enrichment factor of close to 2.6, in approximate agreement with the observed chemical enrichment factors for different elements. The stable isotopes of the groundwater show enrichment by evaporation, which can be accounted for by an average evaporative loss of about 20% compared with the rainfall source of the water. Comparison of the isotopic compositions of groundwater in the Tinwald hot spot and outside it suggest that the irrigation input to recharge is up to 40%. The effects of irrigation return flow are not often described but have considerable management implications, e.g. modelling of nitrate transport through soil/groundwater systems could be highly unrealistic if irrigation return flow effects are disregarded. In addition, too much fertilizer could be applied leading to unnecessary economic cost and greater leaching of nitrate to groundwater if nitrate in irrigation water is not accounted for.

Irrigation return flow also appears to have caused a blending of the nitrates from different sources in the soil as shown by their nitrate isotope compositions. The blended source is dominated by fertiliser which has been transformed by the soil-plant system into organic-N which acts as the important soil N reservoir from which N is mineralised and oxidised during leaching, with effluent contributing to a minor extent. The blending of the different nitrate sources allowed clear identification of the denitrification process in this study. Denitrification enrichment factors of $\varepsilon(^{15}N)$ = -3.0‰, $\varepsilon(^{18}O)$ = -2.1‰ were determined. The nitrate concentration and isotopic compositions were found to be correlated with the DO concentrations because of denitrification, despite most of the groundwaters having DO concentrations greater than the levels at which denitrification can occur. This denitrification imprint is attributed to localised denitrification in fine pores where conditions are reducing, aided by the irrigation return flow process. The implication is that denitrification could be occurring where it is not expected because groundwater DO concentrations are not low.

### Appendix

The fraction of water evaporated ($1$-$f$) is estimated by applying equation 3a from Stewart (1975):

$$\delta = \delta_{end}(1 - f^{\beta}) \tag{A1}$$

where $\delta$ is the enrichment due to evaporation, and $\delta_{end}$ and $\beta$ are given by

$$\delta_{end} = \gamma(\delta_b + 1) - 1 \tag{A2}$$

$$\beta = \frac{1 - \alpha_p \alpha_k (1-h)}{\alpha_p \alpha_k (1-h)} \tag{A3}$$

Here

$$\gamma = \frac{\alpha_p h}{1 - \alpha_p \alpha_k (1-h)} \tag{A4}$$

and $\delta_b$ is the isotopic composition of the atmospheric vapour relative to the initial composition of the water, $\alpha_p$ and $\alpha_k$ are the equilibrium and kinetic fractionation factors respectively between water and vapour, and $h$ is the relative humidity.



Assuming the water composition increases from (-9.43, -65.4) to (-8.58, -61.2) due to evaporation, where the brackets signify ($\delta^2$H‰, $\delta^{18}$O‰), the isotopic enrichment relative to the initial composition of the water is (0.86, 4.5). $\delta_b$ and $h$ were estimated to be (-12, -90) and 70%, respectively. With average temperature of 15ºC, $f$ was calculated to be 0.79 (or fraction of water evaporated *(1-f)* was 0.21).

Chemical enrichment due to evaporation is given by

$$C = C_o/f \qquad (A5)$$

where $C_o$ and $C$ are the initial and final concentrations of the chemical.



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




**Table 1: Sample information, field dissolved oxygen concentration (DO) and isotopes of water and nitrate. The wells are ordered from lowest to highest DO concentrations, and four groups of samples (A to D) are identified for discussion (below).**

| Well No. | ID | Group | Date | Depth (m) | DO mg/L | $\delta^{18}O$ ‰ | $\delta^{2}H$ ‰ | $\delta^{15}N_{NO3}$ ‰ | $\delta^{18}O_{NO3}$ ‰ |
|---|---|---|---|---|---|---|---|---|---|
| K37/0702 | 01 | A | 28/03/18 | 34 | 0.18 | -8.3 | -58.6 | 20.4 | 11.8 |
| K37/3114 | 02 | A | 15/03/18 | 36 | 0.45 | -8.59 | -61.4 | 16.6 | 7.8 |
| K37/1806 | 03 | A | 15/03/18 | 56 | 0.67 | -8.47 | -58.4 | 15.7 | 7.8 |
| K37/2977 | 04 | B | 8/02/18 | 49 | 1.36 | -8.83 | -64.2 | 7 | 3.1 |
| K37/1014 | 05 | B | 22/02/18 | 10 | 1.68 | -8.75 | -62.9 | 7.2 | 3.3 |
| K37/0819 | 06 | A | 22/02/18 | 40 | 2.44 | -8.5 | -60.9 | 20 | 6.5 |
| K37/0147 | 07 | | 7/02/18 | 9.8 | 2.68 | -8.18 | -60.2 | 5 | 0.7 |
| K37/1862 | 08 | B | 15/03/18 | 36 | 2.95 | -8.84 | -62.1 | 8 | 3.9 |
| K37/0336 | 09 | B | 9/04/18 | 7 | 3.39 | -8.74 | -61.6 | 8.7 | 4 |
| K37/3052 | 10 | | 22/02/18 | 15 | 4.11 | -8.8 | -63.5 | 4.2 | 0.2 |
| K37/1012 | 11 | | 15/03/18 | 31 | 6.34 | -8.38 | -58.7 | 6 | 2.8 |
| BY21/0125 | 12 | | 28/03/18 | 30 | 6.73 | -8.53 | -59.8 | 3.5 | 0.3 |
| BY21/0184 | 13 | | 7/02/18 | 11 | 7.76 | -8.18 | -59.1 | 4.5 | 1.2 |
| K37/0088 | 14 | | 12/03/18 | 10 | 8.05 | -8.48 | -61.4 | 5.9 | 1.8 |
| K37/1972 | 15 | C | 7/02/18 | 20 | 8.29 | -8.35 | -59.3 | 3.7 | 1.1 |
| K37/0961 | 16 | | 8/02/18 | 59 | 8.37 | -8.52 | -60.6 | 4.1 | 1 |
| K37/0697 | 17 | | 14/03/18 | 22 | 8.65 | -8.6 | -60.5 | 3.3 | 0.1 |
| K37/2347 | 18 | | 12/03/18 | 60 | 8.76 | -8.57 | -62 | 2.5 | -0.1 |
| K37/1767 | 19 | | 28/03/18 | 30 | 9.18 | -8.49 | -59.6 | 2.2 | -0.4 |
| K37/2301 | 20 | | 9/04/18 | 25 | 9.25 | -8.76 | -61.5 | 4.8 | 1 |
| K37/3049 | 21 | | 22/02/18 | 15 | 9.36 | -7.71 | -54.4 | 1.8 | -1.2 |
| K37/1807 | 22 | | 14/03/18 | 24 | 9.39 | -8.56 | -62 | 3 | -0.3 |
| K37/0968 | 23 | C | 10/04/18 | 10 | 9.54 | -8.19 | -60.2 | 2.6 | -0.5 |
| K37/1479 | 24 | | 9/02/18 | 73 | 9.6 | -8.64 | -61.3 | 2.2 | -0.2 |
| K37/1603 | 25 | D | 9/02/18 | 63 | 9.6 | -8.8 | -63.9 | 2.2 | -0.8 |
| BY21/0306 | 26 | | 17/04/18 | 14 | 9.68 | -8.83 | -63.5 | 3.8 | 0.8 |
| BY21/0307 | 27 | C | 17/04/18 | 13 | 9.85 | -8.47 | -61 | 3.3 | -0.5 |
| K37/1661 | 28 | C | 8/02/18 | 11 | 9.9 | -8.36 | -59.7 | 3.7 | -0.1 |
| K37/1939 | 29 | D | 9/04/18 | 39 | 10.3 | -9.17 | -64.3 | 1.7 | -1.1 |
| K37/3146 | 30 | D | 14/03/18 | 54 | 10.3 | -8.94 | -64.4 | 1.8 | -0.8 |
| K37/0502 | 31 | | 8/02/18 | 24 | 10.4 | -8.47 | -61.6 | 4.5 | 1 |
| K37/0685 | 32 | | 14/03/18 | 18 | 11.6 | -8.51 | -60.2 | 3.5 | 0.6 |
| BY20/0151 | 33 | | 9/04/18 | 45 | 11.8 | -8.66 | -61.3 | 2.2 | -0.9 |





**Table 2: Major ion chemistry of water samples.**

| ID No. | Group | DO mg/L | Na mg/L | K mg/L | Ca mg/L | Mg mg/L | Cl mg/L | SO$_4$ mg/L | HCO$_3$ mg/L | NO$_3$ mg/L |
|--------|-------|------|------|------|------|------|------|------|------|------|
| 01 | A | 0.18 | 11.9 | 1.02 | 18.8 | 6.6 | 18 | 11.6 | 78 | 0.064 |
| 02 | A | 0.45 | 15.5 | 1.49 | 27 | 10.5 | 18.1 | 17.5 | 102 | 5.8 |
| 03 | A | 0.67 | 13.2 | 1.41 | 23 | 8.5 | 17.6 | 15.3 | 92 | 2.5 |
| 04 | B | 1.36 | 11.1 | 1.02 | 18.8 | 7.5 | 12.1 | 13.6 | 76 | 3 |
| 05 | B | 1.68 | 18.5 | 1.25 | 29 | 10.4 | 17.7 | 35 | 97 | 4.3 |
| 06 | A | 2.44 | 12.5 | 1.08 | 20 | 7.6 | 17.1 | 19 | 76 | 1.43 |
| 07 | | 2.68 | 15.2 | 1.71 | 35 | 8.2 | 20 | 27 | 75 | 11.8 |
| 08 | B | 2.95 | 15.7 | 1.39 | 24 | 9 | 17 | 25 | 85 | 4.5 |
| 09 | B | 3.39 | 18.4 | 1.32 | 28 | 9.4 | 18.1 | 34 | 102 | 3.9 |
| 10 | | 4.11 | 16.7 | 1.74 | 29 | 9.3 | 16.4 | 27 | 64 | 12.9 |
| 11 | | 6.34 | 18.1 | 1.64 | 27 | 10 | 26 | 26 | 57 | 10.9 |
| 12 | | 6.73 | 14.8 | 1.47 | 29 | 10.3 | 17.2 | 17.1 | 78 | 13.1 |
| 13 | | 7.76 | 15.9 | 1.71 | 24 | 8.3 | 20 | 29 | 54 | 7.7 |
| 14 | | 8.05 | 17.2 | 1.71 | 30 | 9.7 | 21 | 22 | 64 | 13 |
| 15 | C | 8.29 | 17.7 | 1.28 | 35 | 12.4 | 26 | 36 | 55 | 17.4 |
| 16 | | 8.37 | 14.7 | 1.59 | 30 | 9.4 | 19.3 | 22 | 66 | 14 |
| 17 | | 8.65 | 16.5 | 1.37 | 24 | 8 | 17.6 | 24 | 60 | 9.2 |
| 18 | | 8.76 | 16 | 1.5 | 28 | 9.6 | 16.3 | 21 | 57 | 13.9 |
| 19 | | 9.18 | 17.9 | 1.63 | 33 | 11.2 | 21 | 25 | 71 | 15.7 |
| 20 | | 9.25 | 14.4 | 1.39 | 30 | 10 | 15.3 | 22 | 61 | 17.1 |
| 21 | | 9.36 | 14.9 | 1.53 | 29 | 9 | 18.5 | 29 | 69 | 7.7 |
| 22 | | 9.39 | 17.5 | 1.77 | 36 | 11 | 19.6 | 26 | 60 | 21 |
| 23 | C | 9.54 | 18.4 | 1.56 | 31 | 9.8 | 23 | 30 | 63 | 15 |
| 24 | | 9.6 | 14.5 | 1.6 | 32 | 9.3 | 14.9 | 18.4 | 49 | 22 |
| 25 | D | 9.6 | 10.3 | 1.19 | 16.9 | 5.3 | 9.2 | 5 | 48 | 9.3 |
| 26 | | 9.68 | 15.2 | 1.52 | 32 | 8.8 | 21 | 27 | 82 | 9.5 |
| 27 | C | 9.85 | 21 | 2.3 | 43 | 11.4 | 23 | 39 | 53 | 26 |
| 28 | C | 9.9 | 18 | 1.72 | 38 | 11.9 | 22 | 33 | 53 | 25 |
| 29 | D | 10.3 | 9.2 | 1.1 | 14.9 | 4.1 | 6 | 5.8 | 49 | 6.7 |
| 30 | D | 10.3 | 9.5 | 1.1 | 14 | 4.4 | 7.1 | 3.8 | 49 | 7.3 |
| 31 | | 10.4 | 15.3 | 1.63 | 35 | 10.5 | 13.2 | 23 | 77 | 18.5 |
| 32 | | 11.6 | 17.7 | 1.77 | 34 | 10.5 | 21 | 27 | 77 | 12.8 |
| 33 | | 11.8 | 14.8 | 1.5 | 28 | 7.9 | 13.4 | 18 | 61 | 16.8 |






**Table 3: Results of calculations of the $\delta^{15}N$ or $\delta^{18}O$ values of nitrate affected by denitrification. The initial nitrate concentration was 19 mg/L, and the $\delta_o$ values were $\delta^{15}N$ = 3.1‰, $\delta^{18}O_{NO3}$ = 0.0‰. The enrichment factors used were $\varepsilon(^{15}N)$ = -3.0‰, $\varepsilon(^{18}O)$ = -2.1‰.**

| f | $NO_3$ | Ln ($NO_3$) | $\delta^{15}N$ | $\delta^{18}O_{NO3}$ |
|---|---|---|---|---|
|  | mg/L |  | ‰ | ‰ |
| 1.0 | 19 | 2.94 | 3.1 | 0.0 |
| 0.7 | 13.3 | 2.59 | 4.2 | 0.7 |
| 0.5 | 9.5 | 2.25 | 5.2 | 1.4 |
| 0.3 | 6.65 | 1.89 | 6.7 | 2.5 |
| 0.2 | 3.8 | 1.34 | 7.9 | 3.3 |
| 0.1 | 1.9 | 0.64 | 10.0 | 4.7 |
| 0.04 | 0.76 | -0.27 | 12.8 | 6.6 |
| 0.02 | 0.38 | -0.97 | 14.8 | 8.0 |
| 0.01 | 0.19 | -1.66 | 16.9 | 9.4 |
| 0.004 | 0.076 | -2.58 | 19.7 | 11.3 |


**Table 4: Average isotopic and chemical concentrations of Groups C and D.**

| Quantity | Group D | Group C | Factor |
|---|---|---|---|
| Cl (mg/L) | 7.4 | 23.5 | 3.2 |
| $SO_4$ (mg/L) | 4.9 | 34.5 | 7.0 |
| $HCO_3$ (mg/L) | 48.7 | 56 | 1.2 |
| $NO_3$ (mg/L) | 7.8 | 20.9 | 2.7 |
| Na (mg/L) | 9.7 | 18.8 | 1.9 |
| K (mg/L) | 1.13 | 1.72 | 1.5 |
| Ca (mg/L) | 15.3 | 36.8 | 2.4 |
| Mg (mg/L) | 4.6 | 11.4 | 2.5 |
| Mean |  |  | 2.8 |
| sd |  |  | 1.7 |
|  |  |  | Difference |
| $\delta^{18}O$ (‰) | -8.97 | -8.34 | 0.63 |
| $\delta^2H$ (‰) | -64.2 | -60.1 | 4.1 |



**Captions**

**Figure 1: a. New Zealand map showing locations of Tinwald and other areas mentioned in the text. b. Tinwald study area with simplified land usage (Agribase, 2016). Base map containing road and stream information © LINZ (2019).**

**Figure 2: Maximum nitrate concentrations in the greater Ashburton area 1990 to 2017. Base map © LINZ (2019).**

**Figure 3: Wells sampled in the Tinwald study area for this study. Base map © LINZ (2019).**

**Figure 4: a). Chloride, b). Sulphate, c). Nitrate concentrations in the Tinwald study area (the smaller dots indicate maximum concentrations measured prior to the 2018 investigation). Base maps © LINZ (2019).**

**Figure 5: Plots of: a. chloride versus water $\delta^{18}O$, b. chloride versus sulphate, and c. chloride versus nitrate concentrations. Groups C (with land surface irrigation recharge) and D (alpine river recharge) are circled. The red arrow shows the predicted effect of evaporation.**

**Figure 6: $\delta^{18}O$ in the Ashburton area at groundwater and surface water sites. Data outside of the Tinwald study area is mean $\delta^{18}O$ from all available measurements, inside the study area $\delta^{18}O$ is the single result from the current sampling. Base map © LINZ (2019).**

**Figure 7: Paired $\delta^2H$ and $\delta^{18}O$ data in the Tinwald study area. The average initial composition of the well waters is indicated by the small red circle. Groups C and D samples are circled.**

**Figure 8: a. Plot of $\delta^{15}N$ versus $\delta^{18}ONO3$. Groups A and B are circled, C and D indicated by letters. b, c. Plots of $\delta^{15}N$ and $\delta^{18}ONO3$ versus the natural log of the nitrate concentration.**

**Figure 9: Schematic view of recharge and irrigation return flow in the Tinwald hot spot.**

**Figure 10. Plot of the nitrate isotopic source signatures from several New Zealand studies. Red rectangle – 0.4 m suction samples at Toenepi Catchment (Clague et al., 2015), purple rectangle - stream samples from Harts**
**Creek (Wells et al., 2016), blue rectangle – oxic water samples from Waihora wellfield northwest of Lake Taupo (Stenger et al., 2018), orange double arrows - $\delta^{15}N$ values only from groundwater in the Waimea Plains (Stewart et al., 2011), and oxic waters (Groups C and D) from the present investigation.**

**Figure 11: a. Plot of DO versus nitrate concentration. b, c. Plots of Ln(DO) versus $\delta^{15}N$ and $\delta^{18}O_{NO3}$.**






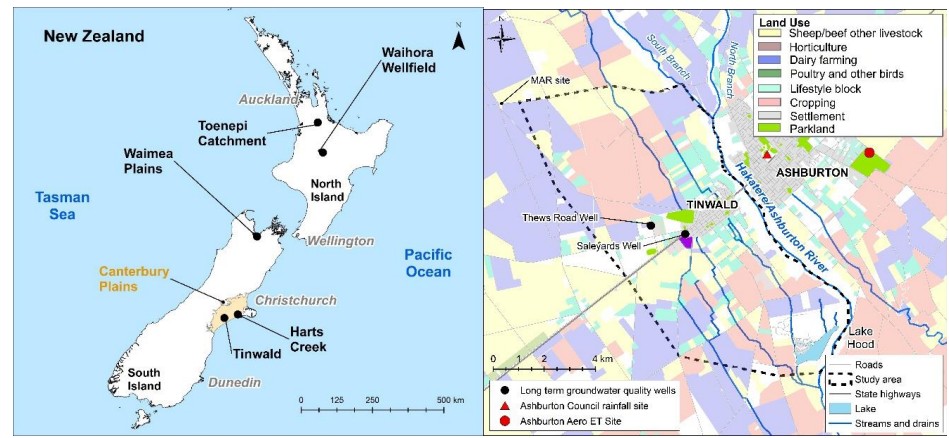

**Figure 1: a. New Zealand map showing locations of Tinwald and other areas mentioned in the text. b. Tinwald study area with simplified land usage (Agribase, 2016). Base map containing road and stream information © LINZ (2019).**


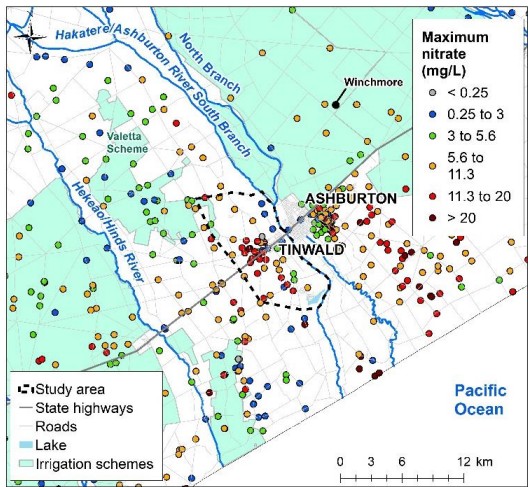

**Figure 2: Maximum nitrate concentrations in the greater Ashburton area 1990 to 2017. Base map © LINZ (2019).**





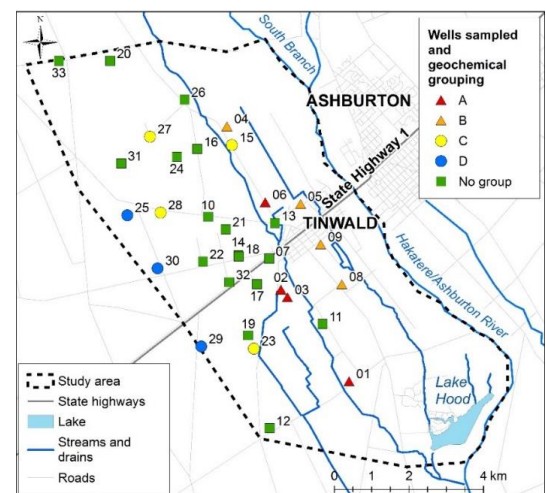


**Figure 3: Wells sampled in the Tinwald study area for this study. Base map © LINZ (2019).**

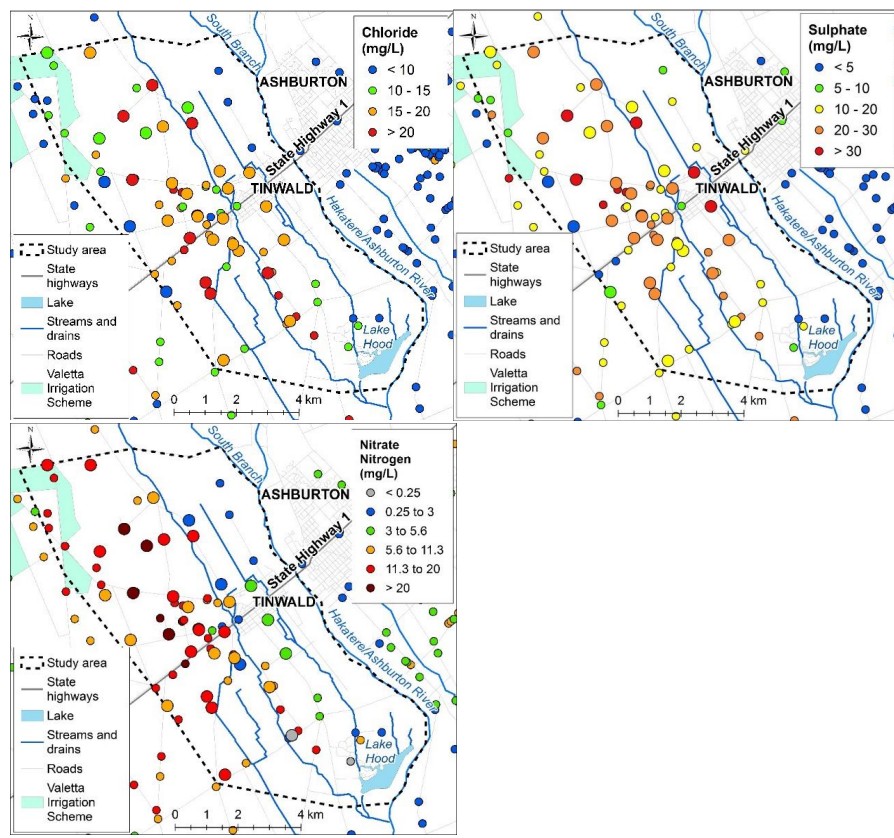

695 **Figure 4: a). Chloride, b). Sulphate, c). Nitrate concentrations in the Tinwald study area (the smaller dots indicate maximum concentrations measured prior to the 2018 investigation). Base maps © LINZ (2019).**





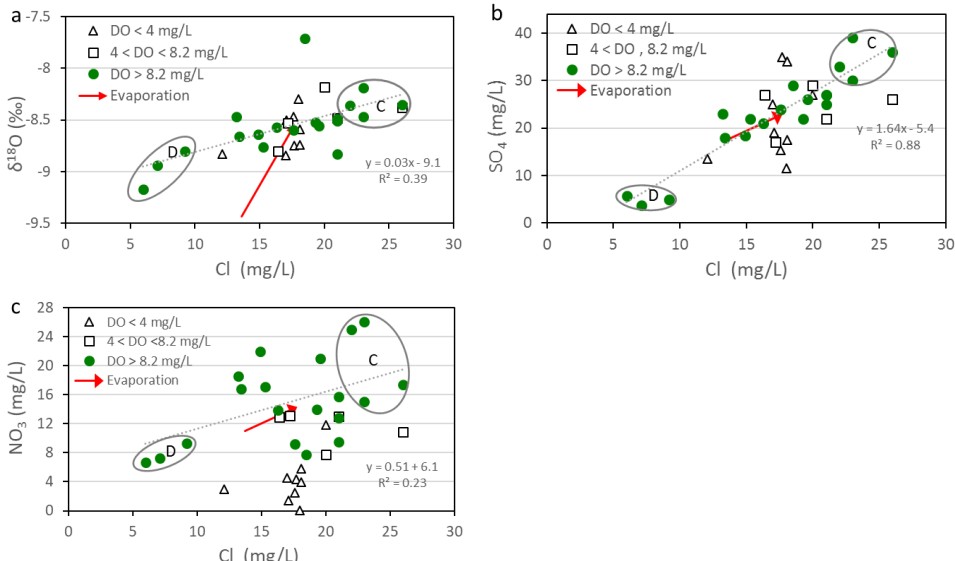

**Figure 5: Plots of: a. chloride versus water δ¹⁸O, b. chloride versus sulphate, and c. chloride versus nitrate concentrations. Groups C (with land surface irrigation recharge) and D (alpine river recharge) are circled. The red arrow shows the predicted effect of evaporation.**

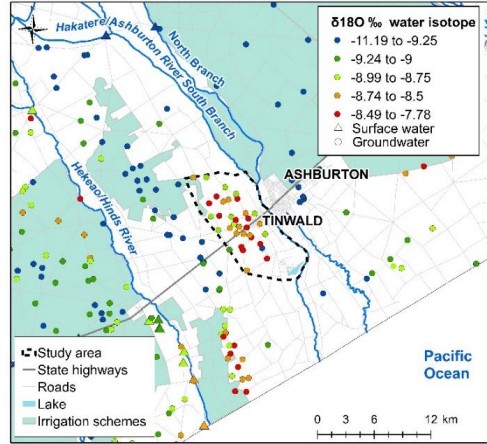

**Figure 6: δ¹⁸O in the Ashburton area at groundwater and surface water sites.  Data outside of the Tinwald study area is mean δ¹⁸O from all available measurements, inside the study area δ¹⁸O is the single result from the current sampling. Base map © LINZ (2019).**





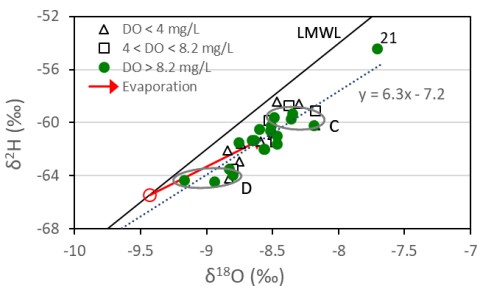

**Figure 7: Paired δ²H and δ¹⁸O data in the Tinwald study area. The average initial composition of the well waters is indicated by the small red circle. Groups C and D samples are circled.**

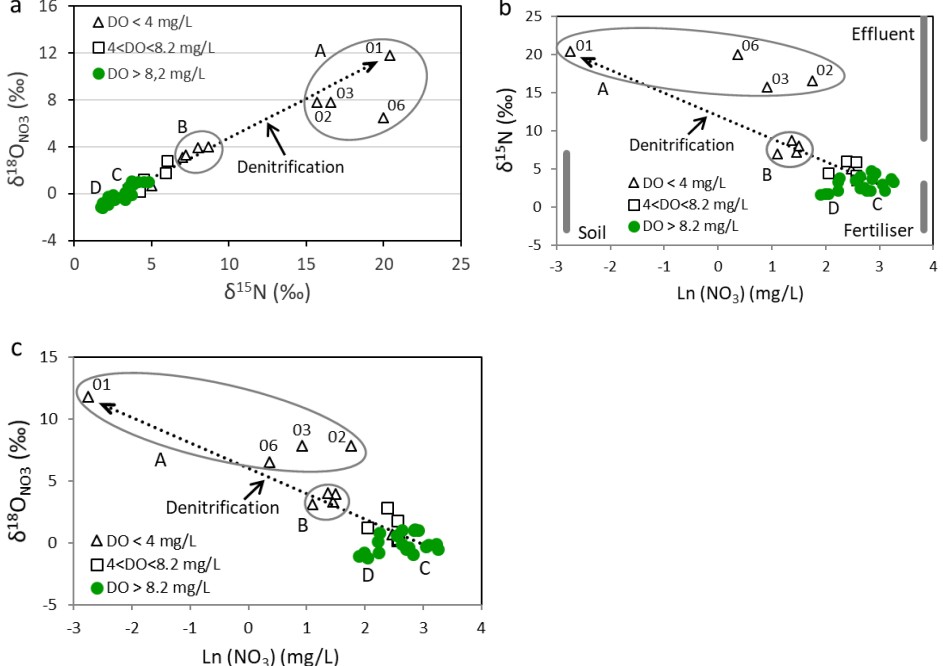

**Figure 8: a. Plot of δ¹⁵N versus δ¹⁸ONO3. Groups A and B are circled, C and D indicated by letters. b, c. Plots of δ¹⁵N and δ¹⁸ONO3 versus the natural log of the nitrate concentration.**





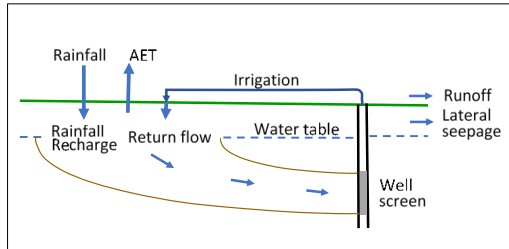

**Figure 9: Schematic view of recharge and irrigation return flow in the Tinwald hot spot.**

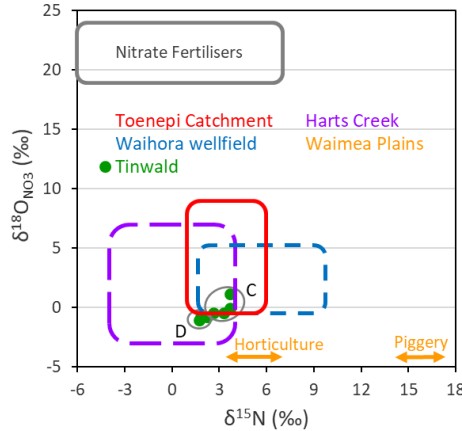

**Figure 10. Plot of the nitrate isotopic source signatures from several New Zealand studies. Red rectangle – 0.4 m suction
samples at Toenepi Catchment (Clague et al., 2015), purple rectangle - stream samples from Harts
Creek (Wells et al., 2016), blue rectangle – oxic water samples from Waihora wellfield northwest of
Lake Taupo (Stenger et al., 2018), orange double arrows - δ15N values only from groundwater in the
Waimea Plains (Stewart et al., 2011), and oxic waters (Groups C and D) from the present investigation.**







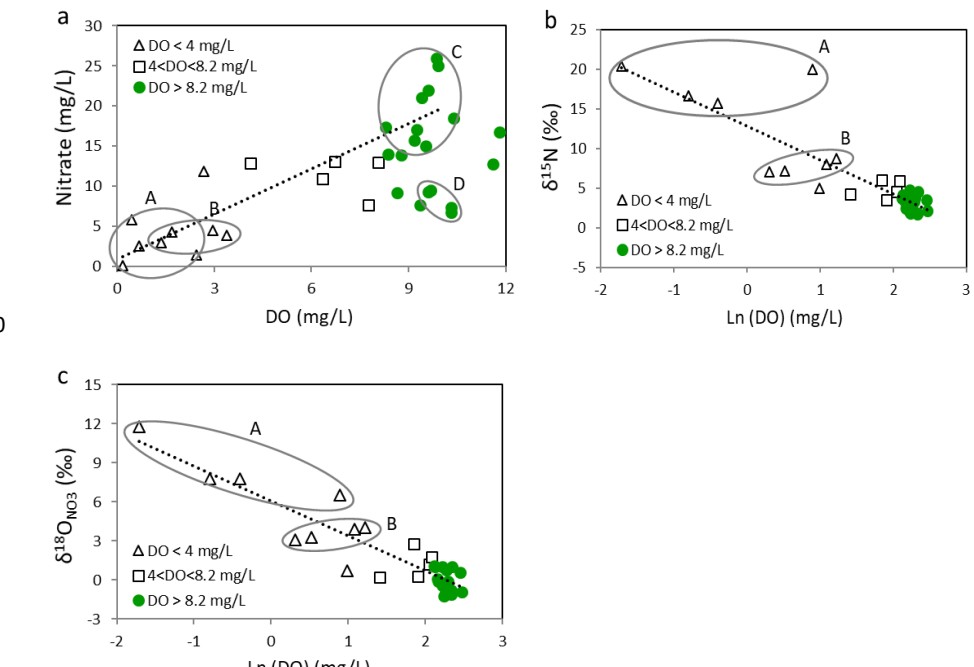

**Figure 11: a. Plot of DO versus nitrate concentration. b, c. Plots of Ln(DO) versus $\delta^{15}N$ and $\delta^{18}O_{NO3}$.**