# Peer review of "Irrigation return flow causing a nitrate hot spot and denitrification imprints in groundwater at Tinwald, New Zealand"

_Hydrology and Earth System Sciences, 2020_

## Referee Comment (RC1) · Anonymous Referee #1 · 31 Mar 2020

The paper considers the contribution by irrigation return flow to high nitrate concentrations in groundwater and points out how this contribution of nitrate needs to be incorporated into assessment of nitrate load limits. The authors also note the occurrence of denitrification processes in a group of wells as indicated by N and O isotopes even though the levels of DO in those wells are not particularly low (up to 4 mg N/ L). This indicates that averaged DO concentrations may not exclude denitrification. The authors suggest denitrification may occur in localised areas with fine pores or reduced microsites even while average DO levels are higher than would be expected for denitrification

to occur.

Principal Criteria Scientific Significance: Good (2) The scientific significance is good – see summary above. There is a discrepancy presented regarding the likely source or location of the denitrification. In the abstract, discussion and conclusions it is suggested that the denitrification may occur in localised areas with fine pores or reduced micro-sites, whereas in the results section (line 272 – 274) it states that the denitrification occurred "most probably within the soil the nitrate was leached from."

Scientific Quality: Good(2) Scientific method and approach are valid and are carried out well. The approach and conclusions could have been improved with the addition of excess N2 data as this would have confirmed denitrification along the flow and added to the conclusions. Presentation Quality: Excellent (1) The paper is written in a clear and readable fashion and is easy to follow. The number and quality of the figures and tables is good and adds to the presentation of information in the paper.

Referee questions – all covered and good.

Specifics The final paragraph in the discussion section (lines 451 – 454) makes an observation about a dual porosity system in the groundwater at Tinwald which is consistent with the proposed site(s) for denitrification. Dual porosity has been studied in these alluvial gravel aquifers by, for example, Dann et al. 2009 and similar papers, and this paper(s) could usefully be incorporated to provide more context and insight to the discussion. Dann, R.L.; Close, M.E.; Flintoft, M.J.; Hector, R.; Barlow, H.; Thomas, S; Francis, G. (2009). Characterization and estimation of hydraulic properties in an alluvial gravel vadose zone. Vadose Zone Journal 8(3): 651-663.

Probable typo – In line 163-164 it gives the precision for the isotope analyses "except for samples below 100 mg/L NO3-N." All samples are below this level so I suspect the units or number is incorrect. Typos: line 505. Denitrification spelt wrong. Line 539. Canterbury spelt wrong Line 545 review spelt wrong.

---

## Referee Comment (RC2) · Anonymous Referee #2 · 7 Apr 2020

This paper looks at the impact of irrigation return waters in areas of intensive agriculture on nitrate concentrations and considers more broadly how we track the sources of nitrate via dual isotopes. The study itself is scientifically sound although I struggled in places with the text. Some of the sections are not very clearly written and the structure could be improved. Specifically, both the introduction and discussion alternate between local and more global observations and the descriptions in parts of sections 4 & 5 are long.

A few clearer statements around the global importance of the work would also improve

its impact.

I have made several comments below that I hope are useful in revising this paper.

Specific comments

Abstract. The abstract is well-written and intelligible without having to refer to the rest of the paper. Having said this, it could be improved by:

1)Adding a sentence at the start to outline the motivation for the study

2)Adding a few key values (there are a fair number of qualitative terms here – high, low, relatively etc). A few specific values would convey more meaning.

Introduction

The introduction sets the scene for the study. For a paper in an international journal such as HESS, it would be appropriate to add a few comments about how New Zealand compares to other intensive agricultural areas globally in terms of the scale of the problem. High nutrient loads are of global interest and this research will have broader interest, so some more comments here are warranted.

The structure could be improved as it alternates between general and area-specific statements. Try to group these more. Some of the description of the issues around Tinwald could be in section 2.

Lines 39-43. What the concentrations of nitrate in the groundwater and the river water?

Lines 44-48. This seems out of place in the description of the local setting. It would be better earlier where you discuss the general importance (especially as it refers to contaminants other than nitrate).

Lines 68-62. These references are a little dated. Can you point to some key recent studies, especially those that deal with the issues of irrigation return flows?

Background

As mentioned above, some of the detail from the introduction (eg the high nitrate, which is also covered in section 2.4) would be better here. This section is also long for the information it contains and probably it could be written more succinctly.

Lines 91-95 (and elsewhere). It would be preferable to use "residence times" not "ages". Also, here and throughout suggest referring to the groundwater not the well (the well's age is when it was installed, which is not what you mean).

Line 125. MAV not defined (I think that it is the maximum WHO limit for nitrate in drinking water?). Is this the best value to use of should the lower NZ limit be used?

Line 126-136. Adding values to the text would get the message across better (rather than the reader having to find them for their self).

Methods

Section 3.1 could use a few more details.

a) Somewhere, the screened intervals should be notes as geochemical data from long-screened production wells conveys different information to that from short-screened monitoring wells.

b) In Table 1, is the depth the mid-screen?

c) Were the wells purged or was sampling done from flowing wells?

d) The Comments on groundwater levels and river flows is not very clear.

Line 147. Field quality?

Lines 148-149. What are the criteria used for these groups?

Results

Section 4.1 is difficult to follow without some more details being reported in the text. At the moment, the reader has to keep looking at the data in the figures or tables to see what is being referred to (especially lines 186-192 and 193 to 200).

[Figure]

Lines 217-223. You do not need the detail of the GMWL as it is well known. You could move the definition of the d excess to the methods.

Lines 220-243. The description of the stable isotopes could be shorter (and clearer). There is a lot of text here to explain a relatively simple concept that samples with isotopic compositions to the right of the MWL are evaporated.

Lines 243-250. There are probably a reasonable number of uncertainties in these calculations (having to estimate the initial isotopic composition of the rainfall that recharges the groundwater, understanding the precise impacts of evaporation etc). Do you have any ides of how those impact these calculations?

Section 4.3. Again, this description is long in places and could be rationalised. The Raleigh equation (line 261) is reported later in the section - it might be better to report how the calculations were done in the methods which would let this section focus on the results (at the very least try to group the descriptions of the calculations and the outcomes more).

Section 5.1. Lines 330-364. Do you need all these calculations? It seems though the Cl mass balance together with the isotopic enrichment defines the recharge % well enough (it is the basis of an often used recharge rate calculation after all). Could you start off with that and then report the results of the water mass balance as support? Also, the infiltration data look to be from a lysimeter, which may be less than total recharge (given that it is probably above the water table). Since you are interested in the chemistry of the recharging water it would be simpler to relate it to a recharge estimate based on the chemistry.

Section 5.2. The other aspect that is often ignored is that the source has to be there (regardless of the isotopic composition). The points made on lines 398-404 are correct but there are a fair few studies where the isotopic compositions point to stores that are not locally present. This more general discussion would be better in the conclusions.

Lines 440-450. It is not clear whether these are general points or some things that may be related to your studies. If it is the latter, is there any evidence that they may apply? You make the point agin in the next paragraph, so something to back it up would be good.

Conclusions

The conclusions are reasonably area specific; however, there are several general points made in the discussion. It is preferable to have the conclusions outline the more general points (after the area-specific conclusions) – then the reader who skips the details gets the message!

Appendix

I am not sure what the rationale is for where the equations are presented. At the moment they are scattered throughout the text and in an appendix. You could look at whether they would all be better in the appendix or split between the appendix and the methods.

Are the calculations in the Appendix the same as those from Gonfiantini (1986: Handbook of Environmental Isotope Geochemistry. Vol.2 the Terrestrial Environment. Elsevier, Amsterdam, 113-186) which are widely used?

Table 4. Standard deviations of the data would be useful

---

## Author Comment (AC2) · 17 Apr 2020

Reply to Anonymous Referee #2 (R2)

R2: This paper looks at the impact of irrigation return waters in areas of intensive agriculture on nitrate concentrations and considers more broadly how we track the sources of nitrate via dual isotopes. The study itself is scientifically sound although I struggled in places with the text. Some of the sections are not very clearly written and the structure could be improved. Specifically, both the introduction and discussion

alternate between local and more global observations and the descriptions in parts of sections 4 & 5 are long. A few clearer statements around the global importance of the work would also improve its impact. I have made several comments below that I hope are useful in revising this paper.

Authors: We thank Referee #2 for helpful and constructive comments on our work.

R2: Specific comments: Abstract. The abstract is well-written and intelligible without having to refer to the rest of the paper. Having said this, it could be improved by: 1) Adding a sentence at the start to outline the motivation for the study 2) Adding a few key values (there are a fair number of qualitative terms here – high, low, relatively etc). A few specific values would convey more meaning.

Authors: We will add sentences to the beginning of the abstract to address points 1) and 2) as follows: "Nitrate concentrations in groundwater have been historically high (N $\geq$ 11.3 mg/L) in an area surrounding Tinwald, Ashburton since at least the mid-1980s. The local community are interested in methods to remediate the high nitrate in groundwater. To do this they need to know where the nitrate is coming from."

R2: Introduction The introduction sets the scene for the study. For a paper in an international journal such as HESS, it would be appropriate to add a few comments about how New Zealand compares to other intensive agricultural areas globally in terms of the scale of the problem. High nutrient loads are of global interest and this research will have broader interest, so some more comments here are warranted. The structure could be improved as it alternates between general and area-specific statements. Try to group these more. Some of the description of the issues around Tinwald could be in section 2.

Authors: We will refer to OECD reports (2013 and 2017), which place New Zealand's intensive agricultural area management in the context of those of OECD member countries: "Eutrophication causing hypoxia and algal blooms, due primarily to agricultural runoff of excess nutrients, is considered the most prevalent water quality problem globally (OECD, 2017). In New Zealand the N balance worsened (i.e. became more positive) more than in any other OECD member country between 1998 and 2009, almost entirely because of expansion and intensification of farming in New Zealand (OECD, 2013). (The N balance is the difference between N inputs to farming systems (fertiliser and livestock manure) and N outputs (crop and pasture production) - a positive N balance indicates increased potential for N pollution of soil, water and air. In fact, the increase of the positive N balance matched the increase of dairy farming in New Zealand.)"

R2: Lines39-43. What are the concentrations of nitrate in the groundwater and the river water?

Authors: Groundwater (N $\geq$ 11.3 mg/L) and alpine river water (N < 1 mg/L). These numbers will be added to the text.

R2: Lines 44-48. This seems out of place in the description of the local setting. It would be better earlier where you discuss the general importance (especially as it refers to contaminants other than nitrate).

Authors: We will move this to later in the paper, but it can't go earlier because the term 'irrigation return flow' needs to be defined first.

R2: Lines 68-62. These references are a little dated. Can you point to some key recent studies, especially those that deal with the issues of irrigation return flows?

Authors: More recent references including two dealing with irrigation return flow as well as nitrate isotopes have been added (i.e. Wexler et al., 2014, Park et al., 2018, Spalding et al., 2019).

R2: Background As mentioned above, some of the detail from the introduction (eg the high nitrate, which is also covered in section 2.4) would be better here. This section is also long for the information it contains and probably it could be written more succinctly.

Authors: We will make these changes.

R2: Lines 91-95 (and elsewhere). It would be preferable to use "residence times" not "ages". Also, here and throughout suggest referring to the groundwater not the well (the well's age is when it was installed, which is not what you mean).

Authors: Agreed

R2: Line 125. MAV not defined (I think that it is the maximum WHO limit for nitrate in drinking water?). Is this the best value to use of should the lower NZ limit be used?

Authors: MAV is now defined in the introduction. We prefer to use MAV, but either limit could be used.

R2: Line 126-136. Adding values to the text would get the message across better (rather than the reader having to find them for their self).

Authors: Agreed, will add values. (This problem is partly due to HESS's requirement that the figures be placed at the end, instead of at the right places in the text.)

R2: Methods Section 3.1 could use a few more details. a) Somewhere, the screened intervals should be notes as geochemical data from long screened production wells conveys different information to that from short-screened monitoring wells. b) In Table 1, is the depth the mid-screen? c) Were the wells purged or was sampling done from flowing wells? d) The Comments on groundwater levels and river flows is not very clear.

Authors: a) Agreed, we will add screened intervals to Table 1 (52% of wells had short screens (average 2 m length), 21% had long screens (average 10 m length) and 27% had no screens). b) The depths given were total depths, but we will use mid-screen depths instead where there were screens and total depths where there were no screens. c) Field measurements had stabilised before sampling for all wells. 25 wells were purged of at least three well casing volumes before sampling, the 8 remaining wells were sampled by low flow methods (pumps were lowered into the wells and water was sampled after pipes were purged by three pipe volumes). d) This has been

rephrased.

R2: Line 147. Field quality?

Authors: No, field quantity (quality means how good (the measurements) are, quantity means what the values are.)

R2: Lines 148-149. What are the criteria used for these groups? Results Section 4.1 is difiñĄcult to follow without some more details being reported in the text. At the moment, the reader has to keep looking at the data in the ïñĄgures or tables to see what is being referred to (especially lines 186-192 and 193 to 200).

Authors: Groups A and B have low DO values (< 4 mg/L) with A having high $\delta$15N (> 15‰ and B moderate $\delta$15N (7-9‰. Groups C and D have high DO (> 8.2 mg/L) with C having the highest and D the lowest Cl and SO4 concentrations. This description will be added to Section 3. The groups are clearly identified in Tables 1 and 2 and in many of the figures. (It is unfortunate that HESS requires that tables and figures be placed at the end of the paper, because it makes the reviewer's task so much more difficult and time-consuming.) Putting in so much detail that no reference to the tables and figures at all is needed would overburden the text.

R2: Lines 217-223. You do not need the detail of the GMWL as it is well known. You could move the deïñĄnition of the d excess to the methods.

Authors: This will be reduced.

R2: Lines 220-243. The description of the stable isotopes could be shorter (and clearer). There is a lot of text here to explain a relatively simple concept that samples with isotopic compositions to the right of the MWL are evaporated.

Authors: Ok

R2: Lines 243-250. There are probably a reasonable number of uncertainties in these calculations (having to estimate the initial isotopic composition of the rainfall that

recharges the groundwater, understanding the precise impacts of evaporation etc). Do you have any ideas of how those impact these calculations?

Authors: This calculation is considered approximate because some important quantities have been estimated rather than being measured. An estimate of error will be made.

R2: Section 4.3. Again, this description is long in places and could be rationalised. The Raleigh equation (line 261) is reported later in the section - it might be better to report how the calculations were done in the methods which would let this section focus on the results (at the very least try to group the descriptions of the calculations and the outcomes more).

Authors: Ok

R2: Section 5.1. Lines 330-364. Do you need all these calculations? It seems though the Cl mass balance together with the isotopic enrichment defines the recharge % well enough (it is the basis of an often-used recharge rate calculation after all). Could you start off with that and then report the results of the water mass balance as support? Also, the infiltration data look to be from a lysimeter, which may be less than total recharge (given that it is probably above the water table). Since you are interested in the chemistry of the recharging water it would be simpler to relate it to a recharge estimate based on the chemistry.

Authors: We will revise this section to simplify it in light of the referee's suggestions.

R2: Section 5.2. The other aspect that is often ignored is that the source has to be there (regardless of the isotopic composition). The points made on lines 398-404 are correct but there are a fair few studies where the isotopic compositions point to stores that are not locally present. This more general discussion would be better in the conclusions.

Authors: We will look at putting some of this into the Conclusions to emphasise some more general points.

R2: Lines 440-450. It is not clear whether these are general points or some things that may be related to your studies. If it is the latter, is there any evidence that they may apply? You make the point again in the next paragraph, so something to back it up would be good.

Authors: As pointed out by Referee #1, other studies have shown the presence of at least two pore sizes in Canterbury gravels (Dann et al., 2009).

R2: Conclusions The conclusions are reasonably area specific; however, there are several general points made in the discussion. It is preferable to have the conclusions outline the more general points (after the area-specific conclusions) – then the reader who skips the details gets the message!

Authors: We'll look at revising the conclusions

R2: Appendix I am not sure what the rationale is for where the equations are presented. At the moment they are scattered throughout the text and in an appendix. You could look at whether they would all be better in the appendix or split between the appendix and the methods.

Authors: Will do

R2: Are the calculations in the Appendix the same as those from Gonfiantini (1986: Handbook of Environmental Isotope Geochemistry. Vol.2 the Terrestrial Environment. Elsevier, Amsterdam, 113-186) which are widely used?

Authors: The results would be very similar, but the calculations are based on the measurements and equations given by Stewart (1975), which also has been widely used and cited.

R2: Table 4. Standard deviations of the data would be useful

Authors: Ok

References Dann, R.L.; Close, M.E.; Flintoft, M.J.; Hector, R.; Barlow, H.; Thomas,

S; Francis, G.: Characterization and estimation of hydraulic properties in an alluvial gravel vadose zone. Vadose Zone Journal 8(3): 651-663, 2009. Park, Y., Kim, Y., Park, S-K., Shin, W-J., Lee, K-S.: Water quality impacts of irrigation return flow on stream and groundwater in an intensive agricultural watershed. Science of the Total Environment 630, 859–868, 2018. Spalding, R. F., Hirsh, A. J., Exner, M. E., Little, N. A., Kloppenborg, K. L.: Applicability of the dual isotopes $\delta$15N and $\delta$18O to identify nitrate in groundwater beneath irrigated cropland. Journal of Contaminant Hydrology 220, 128–135, 2019. Wexler, S. K., Goodale, C. L., McGuire, K. J., Bailey, S. W., and Groffman, P. M.: Isotopic signals of summer denitrification in a northern hardwood forested catchment, Proc. Natl. Acad. Sci. U. S. A.,111(46), 16,413–16,418, 2014.

---

## Author Response (AR1)

5 June 2020

Dear Editor,

5    Thank you for your work on this paper. The paper has been extensively revised in light of the referee's comments. Revised responses and an edited version of the paper are attached.

An approximate list of changes include (detail changes are given in the responses):
Abstract: Sentences added
10   1. Introduction: Extensive changes
2. Background: Values added, changes made
3. Methods: Many changes
4. Results: Sections shortened and revised
5. Discussion: Extensive revisions
15   6. Conclusions: Changes made
Appendices 2 and 3 added
References: New references added
Tables 1 and 4 revised

20   Code/Data availability
None

Author contribution
MKS had major responsibility for the paper, PLA-E responsibility for the research and report that the paper was
25   based on

Competing interests
The authors have no competing interests.

30   The report below forms Supplementary Material for this paper:
Aitchison-Earl, P.: Sources of nitrate in groundwater in the Tinwald, Ashburton area. Environment Canterbury Report No. R19/85, 2019. https://api.ecan.govt.nz/TrimPublicAPI/documents/download/3664244

Yours faithfully,
35   Mike Stewart

Reply to Reviewer #1 (R1)

R1: The paper considers the contribution by irrigation return flow to high nitrate concentrations in groundwater and points out how this contribution of nitrate needs to be incorporated into assessment of nitrate load limits. The authors also note the occurrence of denitrification processes in a group of wells as indicated by N and O isotopes even though the levels of DO in those wells are not particularly low (up to 4 mg N/ L). This indicates that averaged DO concentrations may not exclude denitrification. The authors suggest denitrification may occur in localised areas with fine pores or reduced microsites even while average DO levels are higher than would be expected for denitrification to occur.

Authors: We thank Reviewer #1 for constructive evaluation of our work.

R1: Principal Criteria Scientific Significance: Good (2) The scientific significance is good – see summary above.

There is a discrepancy presented regarding the likely source or location of the denitrification. In the abstract, discussion and conclusions it is suggested that the denitrification may occur in localised areas with fine pores or reduced micro-sites, whereas in the results section (line 272 – 274) it states that the denitrification occurred "most probably within the soil the nitrate was leached from."

Authors: An important point. Our initial conception was that such denitrification would have been within the soil before leaching to the groundwater, but further consideration and perusal of the literature suggested that denitrification can occur in fine pores or reduced micro-sites in the vadose zone – groundwater continuum. We have amended the text to be more consistent with the latter explanation.

R1: Scientific Quality: Good(2) Scientific method and approach are valid and are carried out well. The approach and conclusions could have been improved with the addition of excess N2 data as this would have confirmed denitrification along the flow and added to the conclusions.

Authors: We did not have excess N2 data, but agree this could have strengthened the evidence for denitrification.

R1: Presentation Quality: Excellent (1) The paper is written in a clear and readable fashion and is easy to follow. The number and quality of the figures and tables is good and adds to the presentation of information in the paper. Referee questions – all covered and good.

Specifics The final paragraph in the discussion section (lines 451 – 454) makes an observation about a dual porosity system in the groundwater at Tinwald which is consistent with the proposed site(s) for denitrification. Dual porosity has been studied in these alluvial gravel aquifers by, for example, Dann et al. 2009 and similar papers, and this paper(s) could usefully be incorporated to provide more context and insight to the discussion.

Authors: We agree and will add this reference. We think more evidence is needed here.

R1: Probable typo – In line 163-164 it gives the precision for the isotope analyses "except for samples below 100 mg/L NO3-N." All samples are below this level so I suspect the units or number is incorrect.

Authors: Yes, the number should have been 0.1 mg/L (100 µg/L)

R1: Typos: line 505. Denitrification spelt wrong.

Line 539. Canterbury spelt wrong. Line 545 review spelt wrong.

Authors: These have been corrected.

Reference:

Dann, R.L.; Close, M.E.; Flintoft, M.J.; Hector, R.; Barlow, H.; Thomas, S; Francis, G. (2009). Characterization and estimation of hydraulic properties in an alluvial gravel vadose zone. Vadose Zone Journal 8(3): 651-663.

80 Reply to Anonymous Referee #2 (R2)

R2: This paper looks at the impact of irrigation return waters in areas of intensive agriculture on nitrate concentrations and considers more broadly how we track the sources of nitrate via dual isotopes. The study itself is scientifically sound although I struggled in places with the text. Some of the sections are not very clearly written and the structure could be improved.

85 Specifically, both the introduction and discussion alternate between local and more global observations and the descriptions in parts of sections 4 & 5 are long. A few clearer statements around the global importance of the work would also improve its impact. I have made several comments below that I hope are useful in revising this paper.

Authors: We thank Referee #2 for helpful and constructive comments on our work.

R2: Specific comments: Abstract. The abstract is well-written and intelligible without having to refer to the rest
90 of the paper. Having said this, it could be improved by:

1) Adding a sentence at the start to outline the motivation for the study

2) Adding a few key values (there are a fair number of qualitative terms here – high, low, relatively etc). A few specific values would convey more meaning.

Authors: We added sentences to the beginning of the abstract to address points 1) and 2) as follows:

95 "Nitrate concentrations in groundwater have been historically high (N ≥ 11.3 mg/L) in an area surrounding Tinwald, Ashburton since at least the mid-1980s. The local community are interested in methods to remediate the high nitrate in groundwater. To do this they need to know where the nitrate is coming from."

R2: Introduction The introduction sets the scene for the study. For a paper in an international journal such as HESS, it would be appropriate to add a few comments about how New Zealand compares to other intensive
100 agricultural areas globally in terms of the scale of the problem. High nutrient loads are of global interest and this research will have broader interest, so some more comments here are warranted. The structure could be improved as it alternates between general and area-specific statements. Try to group these more. Some of the description of the issues around Tinwald could be in section 2.

Authors: We have referred to OECD reports (2013 and 2017), which place New Zealand's intensive agricultural
105 area management in the context of those of OECD member countries:

"Eutrophication causing hypoxia and algal blooms, due primarily to agricultural runoff of excess nutrients, is considered the most prevalent water quality problem globally (OECD, 2017). In New Zealand the N balance worsened (i.e. became more positive) more than in any other OECD member country between 1998 and 2009, almost entirely because of expansion and intensification of farming in New Zealand (OECD, 2013). (The N
110 balance is the difference between N inputs to farming systems (fertiliser and livestock manure) and N outputs (crop and pasture production) - a positive N balance indicates increased potential for N pollution of soil, water and air. In fact, the increase of the positive N balance matched the increase of dairy farming in New Zealand.)"

Some description of issues around Tinwald has been moved to Section2.

R2: Lines39-43. What are the concentrations of nitrate in the groundwater and the river water?

115 Authors: Groundwater (N ≥ 11.3 mg/L) and alpine river water (N < 1 mg/L). These numbers will be added to the text.

R2: Lines 44-48. This seems out of place in the description of the local setting. It would be better earlier where you discuss the general importance (especially as it refers to contaminants other than nitrate).

Authors: We couldn't move this earlier because the term 'irrigation return flow' needed to be defined first.

120 R2: Lines 68-62. These references are a little dated. Can you point to some key recent studies, especially those that deal with the issues of irrigation return flows?

Authors: More recent references including two dealing with irrigation return flow as well as nitrate isotopes have been added (i.e. Wexler et al., 2014, Park et al., 2018, Spalding et al., 2019).

R2: Background As mentioned above, some of the detail from the introduction (eg the high nitrate, which is also covered in section 2.4) would be better here. This section is also long for the information it contains and probably it could be written more succinctly.

Authors: We have removed some of the detail from the introduction because it is given here.

R2: Lines 91-95 (and elsewhere). It would be preferable to use "residence times" not "ages". Also, here and throughout suggest referring to the groundwater not the well (the well's age is when it was installed, which is not what you mean).

Authors: These changes have been made.

R2: Line 125. MAV not defined (I think that it is the maximum WHO limit for nitrate in drinking water?). Is this the best value to use of should the lower NZ limit be used?

Authors: MAV is now defined in the introduction. We prefer to use MAV, but either limit could be used.

R2: Line 126-136. Adding values to the text would get the message across better (rather than the reader having to find them for their self).

Authors: Values have been added to the text.

R2: Methods Section 3.1 could use a few more details. a) Somewhere, the screened intervals should be notes as geochemical data from long screened production wells conveys different information to that from short-screened monitoring wells. b) In Table 1, is the depth the mid-screen? c) Were the wells purged or was sampling done from flowing wells? d) The Comments on groundwater levels and river flows is not very clear.

Authors: a) Screened intervals and mid-screen depths have been added to Table 1. b) The depths given were total depths, but we now use mid-screen depths instead where there were screens, and total depths where there were no screens. c) Field measurements had stabilised before sampling for all wells. 25 wells were purged of at least three well casing volumes before sampling, the 8 remaining wells were sampled by low flow methods (pumps were lowered into the wells and water was sampled after pipes were purged by three pipe volumes). d) This has been rephrased.

R2: Line 147. Field quality?

Authors: No, field quantity (quality means how good (the measurements) are, quantity means what the values are.)

R2: Lines 148-149. What are the criteria used for these groups? Results Section 4.1 is difficult to follow without some more details being reported in the text. At the moment, the reader has to keep looking at the data in the figures or tables to see what is being referred to (especially lines 186-192 and 193 to 200).

Authors: Groups A and B have low DO values (< 4 mg/L) with A having high $\delta^{15}N$ (> 15‰) and B moderate $\delta^{15}N$ (7-9‰). Groups C and D have high DO (> 8.2 mg/L) with C having the highest and D the lowest Cl and $SO_4$ concentrations. This description has been added to Section 3. The groups are clearly identified in Tables 1 and 2 and in many of the figures.

R2: Lines 217-223. You do not need the detail of the GMWL as it is well known. You could move the definition of the d excess to the methods.

Authors: This has been reduced.

R2: Lines 220-243. The description of the stable isotopes could be shorter (and clearer). There is a lot of text here to explain a relatively simple concept that samples with isotopic compositions to the right of the MWL are evaporated.

Authors: This has been rewritten.

R2: Lines 243-250. There are probably a reasonable number of uncertainties in these calculations (having to estimate the initial isotopic composition of the rainfall that recharges the groundwater, understanding the precise impacts of evaporation etc). Do you have any ideas of how those impact these calculations?

Authors: A mistake in this calculation has been corrected, Errors in measured quantities (δ) and estimated quantities (δ_b, h, temperature) do not change the fraction evaporated much when the fraction is small (5%) because of the form of the equation.

R2: Section 4.3. Again, this description is long in places and could be rationalised. The Raleigh equation (line 261) is reported later in the section - it might be better to report how the calculations were done in the methods which would let this section focus on the results (at the very least try to group the descriptions of the calculations and the outcomes more).

Authors: Rewritten – equation in Appendix 2.

R2: Section 5.1. Lines 330-364. Do you need all these calculations? It seems though the Cl mass balance together with the isotopic enrichment defines the recharge % well enough (it is the basis of an often-used recharge rate calculation after all). Could you start off with that and then report the results of the water mass balance as support? Also, the infiltration data look to be from a lysimeter, which may be less than total recharge (given that it is probably above the water table). Since you are interested in the chemistry of the recharging water it would be simpler to relate it to a recharge estimate based on the chemistry.

Authors: This has been rewritten.

R2: Section 5.2. The other aspect that is often ignored is that the source has to be there (regardless of the isotopic composition). The points made on lines 398-404 are correct but there are a fair few studies where the isotopic compositions point to stores that are not locally present. This more general discussion would be better in the conclusions.

Authors: Lines 398-404 are needed here for the logic of this section. A summary of this is already in the Conclusions (Lines 470-474).

R2: Lines 440-450. It is not clear whether these are general points or some things that may be related to your studies. If it is the latter, is there any evidence that they may apply? You make the point again in the next paragraph, so something to back it up would be good.

Authors: Lines 440-450 include one piece of evidence directly relevant to the study (Koba et al., 1997) and two pieces of evidence drawn from the study itself. As pointed out by Referee #1, other studies have shown the presence of at least two pore sizes in Canterbury gravels (Dann et al., 2009). This reference has been added to the paper.

R2: Conclusions The conclusions are reasonably area specific; however, there are several general points made in the discussion. It is preferable to have the conclusions outline the more general points (after the area-specific conclusions) – then the reader who skips the details gets the message!

Authors: The conclusions have been slightly revised.

R2: Appendix I am not sure what the rationale is for where the equations are presented. At the moment they are scattered throughout the text and in an appendix. You could look at whether they would all be better in the appendix or split between the appendix and the methods.

Authors: These equations have been put into Appendices 1-3

R2: Are the calculations in the Appendix the same as those from Gonfiantini (1986: Handbook of Environmental Isotope Geochemistry. Vol.2 the Terrestrial Environment. Elsevier, Amsterdam, 113-186) which are widely used?

Authors: The results would be very similar, but the calculations are based on the measurements and equations given by Stewart (1975), which also has been widely used and cited.

R2: Table 4. Standard deviations of the data would be useful

Authors: Standard deviations have been added to Table 4.

References

[revised manuscript text omitted]